# A new production-based model for estimating emissions and banks of ODSs: Application to HCFC-141b

Helen Walter-Terrinoni[1], John S. Daniel[2], Chelsea R. Thompson[2], Luke M. Western[3]

[1]Trane Technologies, Charlotte North Carolina, U.S.A.
[2]NOAA Chemical Sciences Laboratory, Boulder, CO, USA
[3]School of Chemistry, University of Bristol, Bristol, UK and Center for Sustainability Science and Strategy, Massachusetts Institute of Technology, Cambridge, MA, USA

*Correspondence to*: Helen Walter-Terrinoni (helen.a.walter-terrinoni@outlook.com)

**Abstract.** The Montreal Protocol on Substances that Deplete the Ozone Layer requires that the production of long-lived ozone-depleting substances (ODSs) that are intended for use in emissive applications be phased out. The Protocol does not, however, limit the release to the atmosphere of ODSs that currently exist in applications and equipment. Accounting for emissions from these "banked" ODSs (e.g., in insulating foams) is important for monitoring the success of and compliance with the Protocol, for understanding where further mitigation of ODS emissions might be effective, and for estimating future ozone depletion. Here, we present a new bottom-up model that incorporates existing use and life-cycle information to calculate emissions and banks as well as uncertainties in the quantities. To demonstrate the model, we apply it to 1,1-dichloro-1-fluoroethane (HCFC-141b), a chemical used primarily in foam insulation and whose production is currently being phased out. We calculate global emission trends that are qualitatively similar to those derived from atmospheric measurements for the period from 1990 to 2017. After 2017, our emissions no longer track the observationally based trends through the end of the comparison in 2021. This discrepancy suggests either a growing additional source of emissions that is inconsistent with reported production or a model deficiency that was not apparent before 2017. Our calculations also show that the easily accessible bank will be much smaller in the future than the total bank estimated in other recent work, with implications for the feasibility of recovering banks before the release of HCFC-141b to the atmosphere.

## 1 Introduction

The Montreal Protocol on Substances that Deplete the Ozone Layer entered into force on 1 January 1989 with the stated purpose to "protect the ozone layer" (UNEP, 2025). By effectively controlling the supply (e.g., production, import, export, and destruction) of the classes of halogenated chemicals that have been responsible for the most ozone depletion, the Protocol has led to substantially reduced use and emissions of ozone-depleting substances (ODSs) with large benefits to both stratospheric ozone and climate change (Velders et al., 2007). The Kigali Amendment to the Protocol is expected to substantially reduce

future emissions of hydrofluorocarbons (HFCs), which would lead to large further climate benefits (Velders et al., 2009). At the extreme and somewhat speculative end of the spectrum, it has been suggested that without the Protocol, the Earth might have experienced catastrophic ozone loss across most of the globe by the middle of this century (Newman et al., 2009) along with commensurate negative health effects (Slaper et al., 1996).

Under Article 7 or the Protocol, Parties are required to report the import, export, and production of controlled substances to the Ozone Secretariat every year. By 2021, the Protocol had led to a reduction in reported global ozone-depletion potential (ODP)-weighted production of ODSs of about 99% when compared with the peak production in the late 1980s (UNEP, 2024). Emissions have also dropped substantially from the peak. Emissions of ODSs are not reported, nor are they regulated under

the Protocol; however, global emissions can be estimated from changes in atmospheric concentrations of these controlled substances, such as those measured by global observational networks (Montzka et al., 2015; Prinn et al., 2018). The emissions decline has been somewhat slower than that of production because, for the majority of current uses for ODSs, most emissions occur years to decades after production. This emission lag exists because large quantities of ODSs have been used in applications such as refrigeration and air conditioning (R/AC), insulating foams, and some fire-fighting equipment. ODSs from

fire-fighting equipment are emitted only when the contents in the equipment are intentionally deployed or inadvertently released. Emissions from use as a refrigerant in R/AC applications and insulating foam applications occur during active use of the product, upon failure of the product, or when it is decommissioned at end-of-life, and can continue once the product containing the controlled chemical is in a landfill. The abundances of chemicals residing in applications actively being used are referred to as "active" banks, and those residing in foams or equipment that have already been decommissioned and

landfilled are referred to as "inactive" banks. Without intervention, almost all ODSs in these banks are expected to eventually be emitted into the atmosphere. This eventuality is one of the reasons that the WMO scientific assessments of ozone depletion (e.g., Daniel and Reimann et al. (2022)) quantify the future impact of emissions from banks on both ozone depletion and climate forcing. If there were a desire to try to reduce the amount of banked ODSs that would otherwise enter the atmosphere, knowledge of the types of banks is important, as inactive banks are expected to often be more difficult and expensive to capture

than active banks; furthermore, some active banks (i.e., ODSs used in building insulation) are more expensive to recover than other banks (e.g., ODSs in refrigeration) (Mathis, 2011). An accurate understanding of banks and their magnitudes is also important in assessing whether estimated emissions to the atmosphere are consistent with reported production levels for some ODSs. Such comparisons have important scientific and policy implications for compliance with the Montreal Protocol (Montzka et al., 2018; Chipperfield et al., 2021).


Global bank sizes and future emissions projections have been estimated with at least two different approaches in recent stratospheric ozone assessments and the recent literature. One approach has been to start with a bottom-up estimate of the global bank in 2008 for each long-lived ODS (IPCC/TEAP, 2005) and then to calculate the historical bank for subsequent years by adding annual reported production and subtracting annual emissions estimated from global atmospheric concentration

observations (WMO, 2011, 2014, 2018). The average annual fraction of the bank that is released over the past few years (i.e., 5-7) is calculated and then projected into the future, allowing for future estimates of the bank and emissions. Most recently (WMO, 2022), a Bayesian analysis was performed (Lickley et al., 2022) that attempts to best match the historical emissions of an ODS by allowing actual ODS production to be fit as some factor larger than what was reported (i.e., to account for under-reporting) and then optimizing the fraction of the ODS bank that is released over all years. Both of these approaches have been

used to inform policymakers about the ODS amounts currently in banks and about future emissions if there were no further intervention.

These two approaches have served the Montreal Protocol community well and have provided information that responds to questions related to emissions deviations from projected trends. In particular, they represent straightforward approaches to

estimate future emissions and banks, assuming future annual bank release rates remain the same as they have been over some past time period. However, neither method has been applied in a way to include potential changes in bank release rate patterns over time, which are expected as market segmentation changes and the amount of the chemical in each life-cycle stage changes. Furthermore, neither methodology has been used to estimate how accessible the bank is at any given time should policymakers wish to take additional action to control ODS emissions. Both of these approaches also rely on derived atmospheric emissions

based on chemical concentration observations, which can introduce a bias in the emissions and banks estimates arising from any potential error in the atmospheric lifetime of the chemical as well from systematic errors in the observations.

Product-based approaches are examples of bottom-up methods that represent an alternative way to estimate emissions (e.g., (Gluckman Consulting, 2022; IPCC/TEAP, 2005; Mcculloch et al., 2001; Wang et al., 2015; Zhang et al., 2023). Such

approaches begin with knowledge of the various types of applications and products that contain the chemical of interest and the amount of the chemical used in each of these sectors. Knowledge of the expected emission rate during various product life-cycle stages and of the distribution of residence times in each life-cycle stage allow for estimates to be made of both emissions and bank sizes. Estimating the progression through the various life-cycle stages and emissions at each stage can be informed by policy requirements and commercial trends both regionally and temporally. These bottom-up methods can provide

information about the accessibility of the banked chemical of interest and have the flexibility to consider changes in the bank release rate over the life cycle of applications, as well as bank release rate changes that occur as the type of equipment/application remaining in service (e.g., air conditioning, specific type of foam) evolves over time. One of the advantages of these approaches is that they are generally independent of emissions derived from atmospheric observations and can therefore be used as a basis for determining whether actual emissions are consistent with compliance with existing

regulations. However, associated with the model flexibility and wide range of model inputs are important data gaps that can result in banks and emissions values characterized by large uncertainties and potential biases, which are generally thought to be larger than those associated with top-down emissions estimates. For example, the emission rate of the primary blowing agents from foams is generally very temperature dependent and depends on the thickness of the foam, on the quality of any

facing material covering the foam, and the tightness of any potential cabinet (e.g., in refrigeration) encasing the foam

(Andersons et al., 2021; Bomberg et al., 1994; Christian et al., 1991; Hueppe et al., 2020; Makaveckas et al., 2021; Wilkes et al., 2003; Holcroft, 2022). Because of these large sensitivities and the assumptions that must be made to achieve a global or regional average, detailed error analyses are vital to understanding the robustness of such bottom-up approaches.

HCFC-141b (1,1-dichloro-1-fluoroethane, $CH_3CCl_2F$) is a particularly good compound to use for a bottom-up model such as

will be described here. It has been primarily used as a foam-blowing agent with minor use as a solvent, and was introduced as a closed-cell foam blowing agent (FBA) in response to the phase-out of chlorofluorocarbon-11 (CFC-11). Because it was not produced and used in substantial amounts until the 1990s, there is a nearly complete dataset of reported production (UNEP, 2024) and atmospheric concentration observations (WMO, 2022) throughout its history of use, unlike datasets for many other ODSs that are significantly banked. Commercial uses of HCFC-141b are also well understood globally and regionally, as are

policies and other issues that can affect markets over time. These factors allow for better validation of this methodology and the associated assumptions than could be performed with many other compounds.

Here, we present a bottom-up model that calculates banks in and emissions from foam applications. We apply it to HCFC-141b, incorporating knowledge of markets that use this compound and a nearly complete dataset of its production as reported

to the Montreal Protocol's Ozone Secretariat. We compare the model's emissions to estimates determined from atmospheric measurements, and we compare bank size results to those of previous studies, while providing a more refined prediction of global banks than has been previously available. We also provide an example of regional bank and emission results. The description of the model and underlying data used in it are found in Sect. 2. The results and discussion are found in Sect. 3, and our conclusions regarding the modeling approach, its applications to HCFC-141b, and opportunities for its application to

other compounds are in Sect. 4.

## 2 Methods

The life-cycle stages during which HCFC-141b emissions occur and are calculated by our model are shown in Fig. 1. The rest of this section will describe how our model calculates these emissions for different applications. It is the sum of the emissions over the entire life cycle of each market that represents total emissions at any given time. For each market

$$E_i^{total} = E_i^{production} + E_i^{solvent} + E_i^{install} + E_i^{use} + E_i^{decom} + E_i^{landfill} \tag{1}$$

where $E_i^{total}$ is the total emission of HCFC-141b in year $i$. In the equations that follow, we assume there is only a single market, for simplicity. The stages include: (1) production of HCFC-141b, storage, and transport before sale ($E_i^{production}$); (2) chemical blending, shipment, and storage of blended systems, and foam blowing and installation ($E_i^{install}$); (3) active product use ($E_i^{use}$); (4) product decommissioning ($E_i^{decom}$); and (5) the time after decommissioning when the product is no longer used

and is in its final point of disposition (i.e., landfill)( $E_i^{landfill}$ ). For HCFC-141b, there is also emission associated with its use as a solvent ($E_i^{solvent}$). Tracking the transition of a foam blowing agent through the life-cycle stages and estimating the emissions at each stage requires knowledge of numerous parameters that depend on the specific application and can also depend on the geographic region where consumption occurs. These parameters, and their sometimes-sizable uncertainties, are described in the following sections. Parameter values are based on values in the literature, with reference to experimental

results when available. With our approach, emissions are calculated at the regional level for each type of product and at each life-cycle stage. The regions are defined in Appendix 4 of UNEP (2007) and shown in Figure 2, with the only difference imposed here being that Japan is included with Europe so as to not perform any calculation with a region comprised of a single country. These two regions were combined because their use patterns were relatively similar over the time of their peak use. While fundamental model emission parameters and lifecycle lifetime parameters are generally assumed to be the same in each

region (with exceptions described below), variations in consumption and the relative sizes of markets in different regions can be large and lead to different total regional emissions and bank characteristics. Regional results are combined to provide annual global emissions and banks estimates. In this work, the only regional parameter variation we consider in our primary results is a reduced emission associated with the unique decommissioning policy in Europe after 2002, described further in Sect. 2.4. There are also some parameters available for China (Zhang et al., 2023; Wang et al., 2015; TEAP, 2019) that differ from the

values we use here, and are evaluated for their impacts on emissions in Sect. 3.

# Lifecycle Emissions of HCFC-141b in Refrigerator Insulating Foam

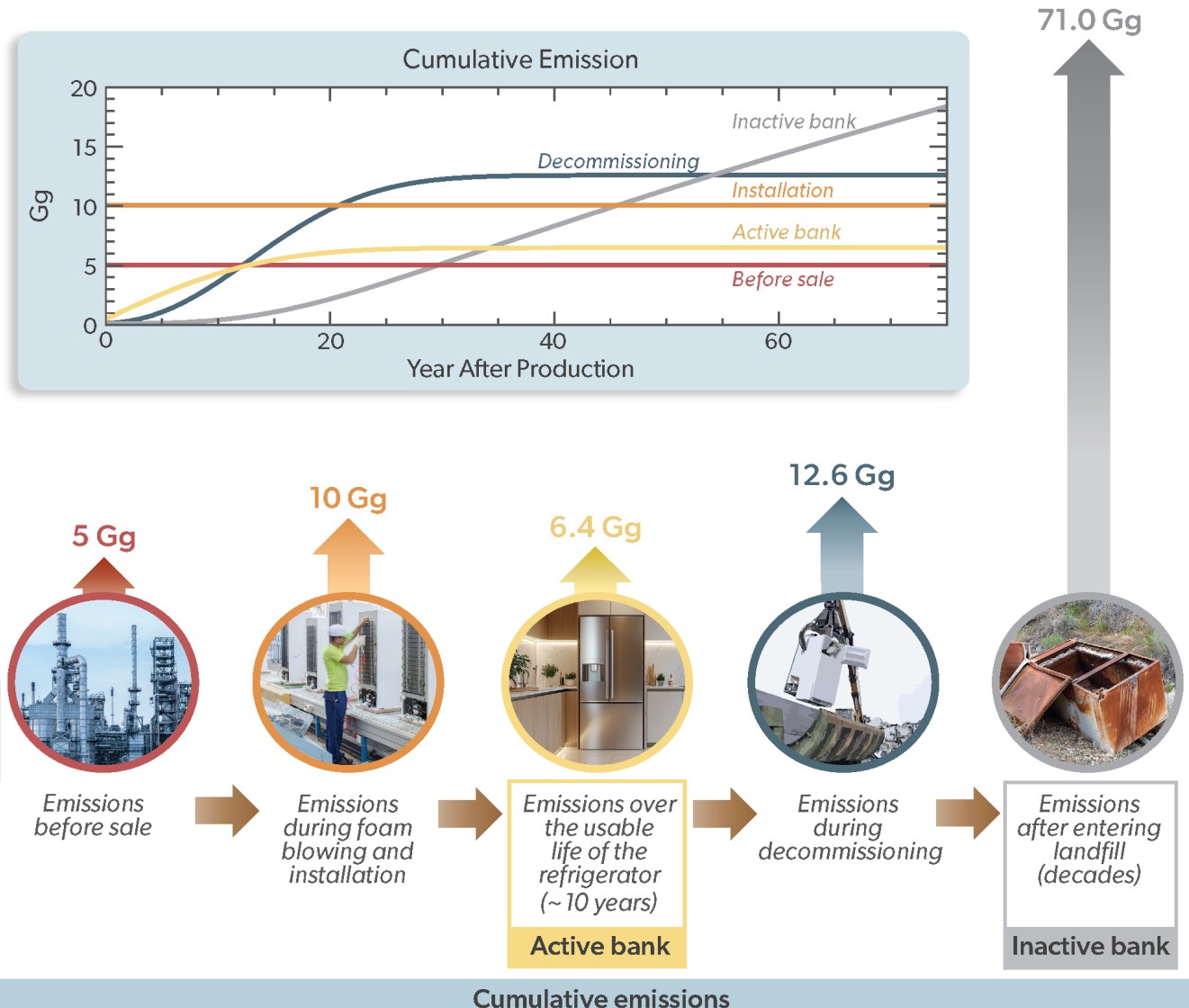

**Figure 1.** Cumulative emissions over time in the life-cycle stages for domestic refrigeration foams as calculated by the model presented in Sect. 2, assuming a hypothetical 100 Gg of reported production for refrigerator foam use in year 0. The emissions before sale, also referred to as "production emissions" in the text, are assumed to not be included in reported production, which is why the sum of all emissions is 105 Gg. The graph shows the first 75 years after production of the 100 Gg, and the numbers in the graphic represent the amount emitted during each life cycle stage after all the HCFC-141b is emitted. The color-coding of the lines is the same as shown in the graphic of the life cycle descriptions and total emissions.

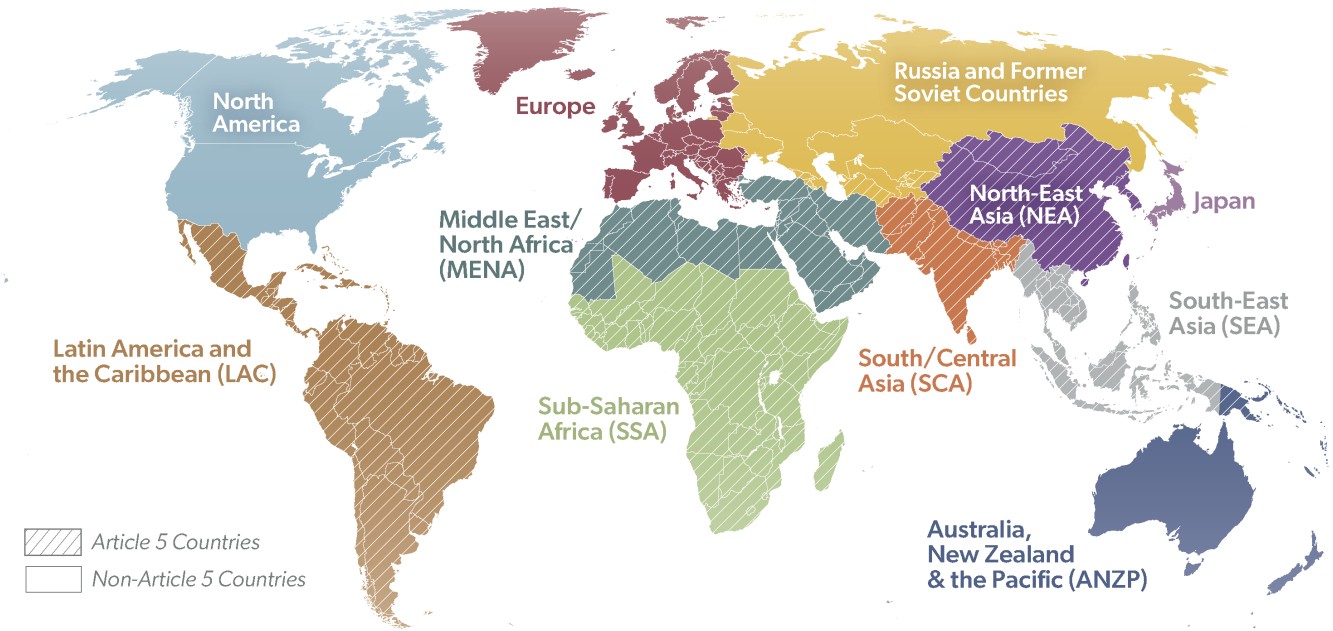

**Figure 2. Regions over which calculations are performed. Also noted is the breakdown of countries into the Article 5 and non-Article 5 categories.**

Parties to the Montreal Protocol annually report chemical production and imported and exported quantities of all controlled ODSs, including HCFC-141b, to the Ozone Secretariat. While compounds are reported as being aggregated by compound groups, the Secretariat has provided specific HCFC-141b values to us with the agreement that no data or results will be shown for any specific country. Calculated consumption, defined as reported production plus imported minus exported chemical is frequently a more appropriate quantity to use for regional calculations as opposed to production, since it better reflects the

local supply of chemicals that are ultimately used in manufacturing and thus is generally better representative of local banks and releases from the foam life cycle. Consumption will form the basis of our calculations. It is worth noting that goods assembled in one location can still be exported to another region with no required reporting of import/export of the ODS contained in the finished product, or even in a chemical blend (e.g., foam system).

For HCFC-141b, annual calculated global consumption is frequently larger than the reported production since the total global quantity of reported imports is higher than reported exports for most years. In the global total, imports should be equal to exports. Thus, if reporting were perfect, by definition consumption and production would be nearly equal in the global sums at least when summing over several years. Even with perfect reporting, individual years might have slightly unequal consumption and production if some quantities were to be attributed to the next year due to supply chain delays between the

timing of export and import. Cumulatively over 1989-2022, consumption is 2.7% higher than production, although differences

can be substantially larger during individual years. We never allow regional consumption to be negative, and thus, after production ends in non-Article 5 (non-A5) Parties, the small reported exports that are in excess of production, are not considered in our calculations. Regionally based consumption over time is shown in **Figure 3**. Global consumption is bimodal, with the earlier peak dominated by non-A5 countries (e.g., Australia, Europe/Japan, Canada, United States), and the latter peak dominated by Article 5 (A5) countries. Global consumption from 2023 onward is assumed to follow a linear trend until is drops to 0 in 2028, consistent with planned phase-out schedules.

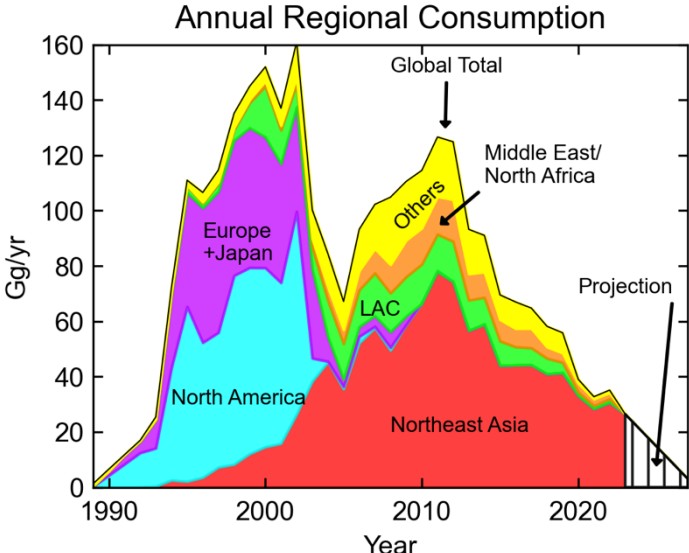

**Figure 3.** Stacked plot of HCFC-141b consumption by region. "MENA" is the Middle East and North Africa region and "LAC" is the Latin America and the Caribbean region. Regions are stacked from bottom to top in descending order of total consumption over the entire historical period. The top of the "Other" region represents the global total consumption. The "Projection" consumption is assumed to be all in the Northeast Asia region.

## 2.1 Emissions before sales and when used as a solvent

The reported production and consumption values do not include losses that occur during the initial chemical production of HCFC-141b and before it is sold for use. These losses can occur while filling containers, drums, tanks, etc., in preparation for sale. They have been estimated by the Technical and Economic Assessment Panel (TEAP) to the Montreal Protocol Parties as varying from 0.9-4% percent of the total production for current "heavily regulated sophisticated plants" to 3-5% in "regulated manufacturing plants" over 1960-1980s, as shown in Tables 2.6 and 2.8 of the Medical and Chemical Technical Options Committee (MCTOC) Assessment Report (MCTOC, 2022). The overall emission factor has been estimated as 4% in IPCC (2019), with an extremely large potential variation across individual facilities. Here, we assume that these losses occuring

before production reporting are 5% of the reported production (4.8% of total production); in our error analysis, this emission factor is assumed to be uniformly distributed between 0 and 10%, with the emissions released to the atmosphere in the same year the production occurred. We do not consider separately any other production that might simply be unreported. Lickley et al. (2022) found emissions were better explained if HCFC-141b production were 12% higher than what was reported. If we let $\varepsilon^{prod}$ be the emission fraction that occurs before sale,

$$E_i^{production} = \varepsilon^{prod} P_i \tag{2}$$

where $P_i$ is the reported production in year $i$.

HCFC-141b has also been used as a solvent. *UNEP* (2003b) estimates that 10% of the produced HCFC-141b was used as a solvent at the time that report was published; more recently, *UNEP* (2019) estimates this has been between 5 and 7.5% from 2011-2018 in A5 countries, while Zhang et al. (2023) has stated that solvent use represented 9.2% of production in 2019 and 8.0% of cumulative production over 2000-2019 in China. We assume that solvent use is 10% of global annual reported production, with a 50% uncertainty on this number (i.e., a uniform range from 5%-15%). Half of the HCFC-141b used as a solvent is assumed to be emitted in the year of being manufactured with the rest emitted in the following year (IPCC, 2006). If $f_s$ is the fraction of reported production of HCFC-141b that is used as a solvent,

$$E_i^{solvent} = 0.5 f_s (P_i + P_{i-1}) \tag{3}$$

**2.2 Emissions during foam blowing agent installation**

Blowing agent releases at the beginning of product life depend on the specific application. For example, foam blowing agent (FBA) releases during foam manufacture from spray foam applied in buildings are thought to be greater than releases during a controlled injection process into a mold during the manufacturing of appliances (Aprahamian and Bowman, 2005). To obtain information about the division of HCFC-141b between the various markets most important for HCFC-141b use and considered here (see Table 1 for list of markets), we rely on the three Foams Technical Options Committee (FTOC) Assessment reports that provide this information (UNEP, 2003a, 2007, 2010). These reports have performed a detailed analysis of the markets for each geographic region. The years covered encompass the majority of the transition period from when HCFC use was primarily in non-A5 countries to a time when the majority of HCFC-141b production occurred in A5 countries. These reports also provide information regarding the fraction of each market that uses HCFCs as opposed to other types of compounds (e.g., HFCs, CFCs, etc.); we used this information of fractional use of HCFCs and further consider only the markets where the HCFC used was HCFC-141b rather than HCFC-142b or HCFC-22. This market information is combined with regional consumption estimates (c.f., **Figure 3**) to determine the amount of HCFC-141b used in each of the 11 markets in each region. We have applied the regional market breakdown to consumption data for all years through 2002 as given by UNEP (2003a), with markets from 2003 through 2007 given by UNEP (2007). Values from 2008 through the end of the calculated time period are

given by UNEP (2010), with the added constraint that all use of HCFC-141b in refrigeration foams is linearly phased out over 2010-2015, with the remaining markets scaled up proportionately. This refrigeration phaseout is meant to approximate the impact of the policy implemented in many countries mandating a transition away from using HCFC-141b in appliances by 2015 as part of their ODS phasedown. If we do not apply this phaseout, the largest annual difference in calculated emission is less than 2 Gg, and it does not affect the discussion in Section 3.

"Foam blowing and installation" emissions referred to in Figure 1 include all emissions that occur during the foaming process as well as any excess emission that occurs through the first year of use. These emissions are meant to be globally appropriate values spanning highly controlled large manufacturing facilities to less controlled activities. This incorporates potential losses from the supply chain after sale of the HCFC-141b and prior to the delivery to the manufacturing plants, such as during shipping and handling, when mixing FBAs into polyol blends, during the manufacturing of the foams, and early losses from foams during the first year after manufacturing. Factors such as the volatility of the chemical, its solubility in the polyol, and the extent to which the foam blowing occurs in a controlled environment are important in determining losses during the foaming process. One study found that, given the boiling point of HCFC-141b (32°C), a 4% emission at the time of foam blowing would be expected even in the contained environment of a refrigerator-like mold (Aprahamian and Bowman, 2005). To account for all sources of emission in this category, we assume 10% release rate for foams installed in molds for domestic refrigeration as well as in polyurethane (PU) pipe-in-pipe, panels, and boardstock products, with higher emissions in other markets. We assume an absolute uncertainty standard deviation of 5% on all emissions associated with installation. For example, if the installation emission fraction is 15%, its uncertainty range is 15%±5%, and all installation emissions are assumed to occur in the year of the HCFC-141b production. Installation emission factors are shown in Table 1 for all markets. If the emission factor for installation is $\varepsilon^{install}$,

$$E_i^{install} = \varepsilon^{install}(1 - f_s)P_i \tag{4}$$

**Table 1.** Assumed release rates of HCFC-141b during foam manufacturing and installation and active use, and parameters used to describe the failure rate of products as a function of time. Uncertainties represent the standard deviation of the probability distribution used in the Monte Carlo analysis; probability distribution functions for installation emissions and emissions during foam use (active bank), are both represented by lognormal distributions; pdfs for the Weibull scale factors and Weibull lifetime terms are represented by Normal distributions. Sources for these values are provided in Table A1.

| Application | Emission During Manufacturing and Installation (Sect. 2.2) | Annual Emissions During Foam Use ($\varepsilon$) (Sect. 2.3) | Failure Rate Parameters for Weibull function: $f(t) = \frac{s}{\tau}\left(\frac{t}{\tau}\right)^{s-1} e^{-(t/\tau)^s}$ (Sect. 2.3, 2.4) |
| --- | --- | --- | --- |
| | | | |

|  |  |  | Weibull Scale Factor, s | Weibull Lifetime Term (Mean Lifetime), τ |
|---|---|---|---|---|
| **Refrigeration** |  |  |  |  |
| Domestic refrigeration | **10%** | **0.5%** | **2.34** | **18.1 (16.0)** |
| Commercial refrigeration | **20%** | **0.5%** | **2.34** | **16.9 (15.0)** |
| Refrigerated containers | **20%** | **0.5%** | **2.34** | **19.7 (17.5)** |
| **Building Construction** |  |  |  |  |
| Continuous panels | **10%** | **0.5%** | **1.97** | **67.6 (60.0)** |
| Discontinuous panels | **10%** | **0.5%** | **1.97** | **67.6 (60.0)** |
| Spray foam | **25%** | **1.5%** | **1.97** | **67.6 (60.0)** |
| PU boardstock | **10%** | **1.0%** | **2.8** | **28.1 (25.0)** |
| **Other Uses** |  |  |  |  |
| PU pipe-in-pipe | **10%** | **0.5%** | **3.0** | **33.6 (30.0)[1]** |
| PU block-pipe | **45%** | **7.5%** | **3.0** | **16.8 (15.0)** |
| PU block foam slab | **20%** | **1.0%** | **3.0** | **16.8 (15.0)** |
| PU integral skin | **40%** | **2.0%** | **3.0** | **11.2 (10.0)** |

[1]Uncertainty is assumed to be 30% as described in Table A1.

**Table 2.** Release rates for emissions that are assumed to not depend on market. Uncertainties represent one standard deviation of the probability distribution used in the Monte Carlo analysis for decommissioning and inactive bank release, which are both represented by lognormal distributions. Emissions before sale and the solvent use are assumed to be described by uniform distributions with the full range represented (*i.e.*, 0-10% and 5-15%, respectively). Sources for these values are provided in Table A2.

| Type of Emission | Magnitude and Timing |
|---|---|
| Before sale (includes loss during production) | 5%±5% (uniform distribution function, i.e., 0%-10%) of reported consumption in year produced |
| During Decommissioning | 15%±15% of decommissioned amount in year of decommissioning |
| From inactive bank (i.e., post-decommissioned bank such as landfill) | 0.50%±0.25% of inactive bank, annually |
| Solvent Use (emitted over 2 years) | 10%±5% (uniform distribution function, i.e., 5%-15%) of reported consumption |

## 2.3 Emissions during product use

Emissions during the operational use of foam products also depend on the particular application and generally occur gradually (e.g., Hueppe et al., 2020; Kirpluks et al., 2022; Paul et al., 2021; Wilkes et al., 2001). Descriptions of how the composition of foams changes with aging have been published that estimate changing concentrations of foam blowing compounds from

270 changes in thermal conductivity (Andersons et al., 2021, 2022; Bomberg et al., 1994; Kirpluks et al., 2023) and/or from measuring the gas composition directly (Modesti et al., 2005; Kirpluks et al., 2023). Measurements have shown great variation in diffusivities of the FBA out of foams, with sensitivity to temperature, foam thickness, presence and quality of any facing material on the foam, and the integrity of any casing around the foam. These variations make it unclear how best to extrapolate individual studies to region-wide values. Therefore, our values, with the exception of the block-and-pipe and pipe-in-pipe

products, are generally consistent (when uncertainties are considered) with bottom-up values adopted in other work (Table A4-1 in UNEP (2003a), Table 7-7 in IPCC/TEAP (2005), Tables 7.6 and 7.7 in IPCC (2006), and Table A4.3 in TEAP (2019)). Our block-and-pipe emission rate estimate is taken from what was used in TEAP, 7.5% (TEAP, 2019), although Table A4.3 in TEAP (2019) mistakenly stated that 75% was the value used. To further complicate matters, some references have used 0.75% for HFC emissions (IPCC/TEAP, 2005; IPCC, 2006). For this work, because the block-and-pipe market is relatively

small for HCFC-141b, what we use for this value is of very small relevance. The largest annual difference in emissions between using 0.75% and 75% is less than 0.7 Gg. Our pipe-in-pipe emission rate of 0.5% is close to the 0.25% in the listed references above, whereas the value quoted in TEAP (2019) was 25% and is thought to be too high. We assume the standard deviation (s.d.) for each emission rates is ±100% of the value used and that the uncertainty follows a lognormal distribution.

The length of time foam products remain in use varies greatly and depends on product type. For example, insulated refrigerated containers exposed to heavy vibration over roads will likely have shorter lifetimes compared to insulating foams installed in buildings. Some foam product lifetimes also vary by region, e.g., as in lifetimes of buildings (Deetman et al., 2020). Here, we calculate emissions during the life cycle stage of active product use by using Weibull survival functions for equipment and buildings to create a probabilistic distribution of the active life stage by foam type (e.g., Aktas and Bilec, 2012; UNFCCC, 2017; Yazici et al., 2014; UNEP, 2023; Gallagher et al., 2014). The fraction of equipment that remains in service as a function of time, $t$, after installation is given by

$$F(t) = \exp\left\{-\left(\frac{t}{\tau}\right)^s\right\}, \tag{5}$$

which is 1 minus the Weibull function's cumulative distribution function. The Weibull function is described by two parameters. One, $s$, governs the general shape of the distribution of decommissioning timing, with smaller values implying statistically more abrupt decommissioning. The second, $\tau$, is related to the length of time the product is used before decommissioning, with the value representing the number of years after being put into service when 63% of the products have been decommissioned. The Weibull parameters are shown in the final two columns of Table 1 for each product type. To account for emission during the active life cycle phase, we assume that leakage emissions from the foam in a particular market is given by

$$\frac{dh}{dt} = -\varepsilon h, \tag{6}$$

so that

$$h(t) = h(0)\exp(-\varepsilon t), \tag{7}$$

where $h(0)$ represents the amount of HCFC-141b in the installed equipment at time 0 and $\varepsilon$ is the annual emission rate. Thus, if we normalize $h(0)$ to be 1, the amount of HCFC-141b mass that remains in active equipment at time $t$ is given by

$$M_{active} = h(t)F(t) = \exp\left\{-\varepsilon t - \left(\frac{t}{\tau}\right)^s\right\} \tag{8}$$

and the cumulative amount of emission that has occurred from the active bank through time $t$ is given by

$$E_{active}(t) = -\int_0^t F(t')\frac{dh}{dt}dt' = \varepsilon \int_0^t \exp\left(-\varepsilon t' - \left(\frac{t'}{\tau}\right)^s\right)dt'. \tag{9}$$

Eq. (9) can be summed over all years prior to derive the emission for any particular year, such that when installation emissions and solvent use is considered,

$$E_i^{use} = (1 - f_s)(1 - \varepsilon_{install})\varepsilon \sum_{j=0}^i P_j \int_{t_i-t_j}^{t_i-t_j+1} \exp\left\{-\epsilon t' - \left(\frac{t'}{\tau}\right)^s\right\}dt' \tag{10}$$

The parameters used in Eqs. (8-10) are shown in Table 1 for each of the different markets.

Figure 4 shows, as examples, the decommissioning functions and emissions for three markets. The solid curves represent the fraction of the installed amount of HCFC-141b remaining in active use as a function of time after installation. Also shown are the cumulative emissions that occurred during use. The difference between the total installed and the sum of cumulative

emissions and remaining active bank represents the fraction of installed HCFC-141b that resided in applications that have reached end-of-life. For example, much more of the HCFC-141b in the "Domestic Refrigeration" market eventually goes to the landfill (>90%) than in the "Spray Foam" market (<50%), since more of the HCFC-141b in spray foam is emitted while the foam is still in use.

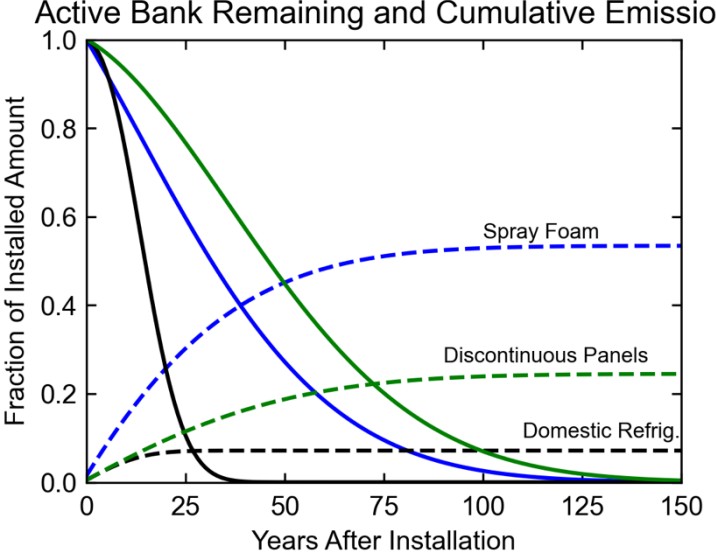

**Figure 4.** Weibull decommissioning functions assumed for three product types: (1) spray foam, (2) discontinuous panels, and (3) and domestic refrigeration foam. These three markets were chosen because their rates of emission from foam and service lifetimes provide a wide range of responses. Solid lines represent the amount of HCFC-141b remaining in the equipment as a fraction of what was installed in year 0. Dashed lines represent the cumulative amount emitted while the product is in use. The domestic refrigeration curve peaks slightly
higher here than in Figure 1 because here, the fractional emission is relative to the amount of HCFC-141b installed, not produced.

## 2.4 Emissions during decommissioning

Significant emissions of HCFC-141b and other similar foam blowing chemicals can occur at the end of life for products that still contain FBA due to the dismantling and disposal processes. In the case of foam products, this primarily results from the partial crushing or shredding of the foam at the time of disposal. Although some foams are recycled or destroyed at this time,
most are transported to landfills as waste. In the United States, most foams are crushed or shredded when they enter the waste stream (ICF International, 2011). Scheutz and Kjeldson (2002) and Kjeldson and Scheutz (2003) measured immediate and short-term release of FBAs of up to 20% for CFC-11 and 28% for HCFC-141b from shredded foams. The amount of FBA emitted during and soon after shredding was highly dependent on the size of the remaining pieces, with courser (finer) particles results in less (more) emission. They did not measure release during the dismantling process. Also, Scheutz et al. (2007)
measured an average release of 24% from shredders typical of United States shredding facilities. Key uncertainties regarding how much FBA gets released at and soon after the time of disposal stem from not knowing how much foam is shredded regionally and globally, and of that amount, how finely are the foams shredded. Values for fractional release as time of disposal

have ranged from negligible (TEAP, 2019) all the way up to 100%. One hundred percent is almost certainly too much release (TEAP, 2005), and recent estimates have ranged from 2% to 20% (TEAP, 2019). We assume a 15% (s.d., 15%, lognormal distribution) release of the FBA that remains in the decommissioned product to describe losses during the dismantling, transport and disposal processes. This is higher than the 5% used in TEAP (2019) for CFC-11 and much lower than the 100% used in McCullough et al. (2001) for CFC-11. The specific choice of this value, for values described by our assumed uncertainty range, matters little to our comparisons or discussions. The decommissioning emissions are applied immediately when the product is retired from service as given by the equations in Sect. 2.3. If $D_i$ is the amount of HCFC-141b in the active bank that is decommissioned in year i, and $e^{decom}$ is the emission rate at the time of decommissioning,

$$E_i^{decom} = \varepsilon^{decom} D_i \tag{11}$$

Europe mandated that FBAs in refrigeration be "recovered for destruction … or for recycling …" during decommissioning of appliances beginning in 2002 (E.U. Regulation No. 2037/2000, Article 16 and Directive 2002/96/EC). While evidence of the extent of compliance is unclear, this would reduce this source of emissions. The decommissioning release rate from domestic appliances in the European market has been reduced to zero from 2002 onward to account for this and the HCFC-141b in any decommissioned application in Europe from 2002 onward is removed from our calculations, not contributing to the future banks or emissions. While this represents an extreme assumption, it matters little to our global bank and emission calculations.

## 2.5 Emissions after decommissioning

After foams are brought to the landfill, and after any initial rapid emission due to crushing or shredding of the foams, FBAs generally continue to be emitted slowly over time. We assume annual release rates of 0.50%±0.50% (TEAP, 2022) of the amount remaining in products after decommissioning. We neglect any potential for anaerobic degradation (Kjeldsen and Scheutz, 2003; Scheutz et al., 2009) so, effectively, the entire inactive bank is eventually released. Whatever is decommissioned and not emitted as in Eq. (11), goes into the inactive bank. If the emission rate of the inactive bank is given by $e^{landfill}$ and $B^{landfill}$ is the size of the inactive bank,

$$E_i^{landfill} = \varepsilon^{landfill} B_i^{landfill} \tag{12}$$

## 2.6 Uncertainty analysis

The model described in the previous sections has a total of 79 input parameters, each with uncertainties associated with it, and each with varying degrees of importance to the calculation of emissions and banks of HCFC-141b. All uncertainties are combined to determine their influence on emissions and banks using a Latin-Hypercube Sampling Monte Carlo approach (e.g., Velders and Daniel, 2014). We perform 5000 simulations to determine the uncertainty ranges in Sect. 3; we find this number is more than sufficient to estimate uncertainty ranges of banks and emissions in a repeatable manner. In each Monte Carlo

simulation, the randomized quantities remain fixed for that entire time series and are not allowed any year-to-year variations. The magnitudes of the uncertainties are shown in Tables 1 and 2, and they, themselves, can also be highly uncertain. We will discuss some of the key uncertainties in Sect. 3. Most uncertainties are assumed to follow lognormal distributions around the primary value. Exceptions are that the amount used as a solvent and the amount emitted as "production emissions" are assumed to follow a uniform distribution with the values listed in Table 2 being the full range of the distribution. Weibull scale factors

and lifetime terms follow a Normal distribution. The uncertainties associated with these variables are assumed to be independent. Market share uncertainties for the 11 different markets are slightly more complicated because all market shares in each simulation must equal 100% and thus are not, by definition, independent. To simulate this, in each Monte Carlo iteration, a 3-step process is carried out: (1) the individual market shares are altered by adding a random number chosen from a standard Normal distribution with a standard deviation equal to 50% of the primary value; (2) all negative market shares are

raised to 0; and (3) then they are all scaled proportionately so they sum to 100%. This approach leads to a slight low bias of the average market share for moderately-sized markets (Figure S1 and Figure S2). It also leads to a somewhat more substantial low bias in the actual standard deviation of the market size distributions for moderately sized market shares (Figure S3).

## 3 Results

### 3.1 Sectoral breakdown of HCFC-141b use

The market segmentation approach described in Sect. 2.2, with information taken from UNEP (2003a), UNEP (2007), and UNEP (2010), yields a market breakdown over time shown in Figure 5. During the first decade of its production, this analysis shows that the majority of HCFC-141b use was in domestic refrigeration foam, spray foam, continuous and discontinuous panels, and boardstock. The applied linear phaseout of refrigeration uses is apparent from 2010 to 2015 with pipe-in-pipe, spray foam, and panel use dominating global markets over the most recent years.

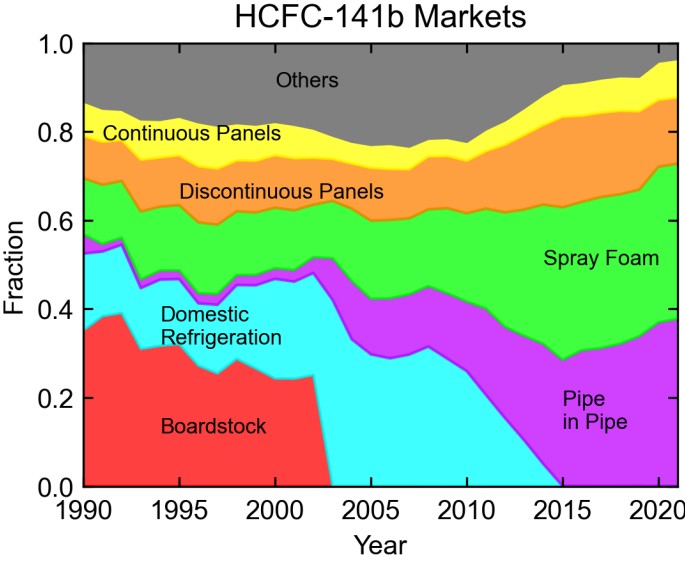

**Figure 5.** Global breakdown of HCFC-141b markets. Values based on regional market fractions and the amount of HCFC-141b used in each market as described in the text combined with regional consumption shown in **Figure 3**.

### 3.2 Annual emissions

The market breakdown is combined with the release parameters of Tables 1 and 2 and annual consumption values (Figure 3) to calculate the emissions and banks of HCFC-141b over time. The calculated total annual emission from the sum of all life cycle stages, all markets, and all regions is shown in Figure 6, with the 1-sigma range shown for all values. The relatively low release rates imply that the contribution of emissions from active and inactive banks changes slowly over time. Higher frequency year-to-year changes are due to rapid emissions, such as those associated with production, emissions from solvent use, and emissions associated with installation of various foams. Due to the large variation in the sizes of the markets as well as in the magnitudes and the uncertainties ascribed to each parameter, there is a large variation in the impact of the uncertainties of each parameter in Tables 1 and 2 on the emission range shown in the figure. To identify the key uncertainties in the calculations, we have performed Monte Carlo calculations for each parameter individually, with all others fixed. The three most important sources of uncertainty are uncertainties in market segmentation, emissions associated with production (and before sale), and amount of solvent use. Each of these tends to change the entire emission curve roughly proportionately over the time period shown, with other uncertainties demonstrating different temporal impacts on emissions (Figure S4). The top 30 uncertainties when averaged through 2024 are shown in Figure S5 and can provide insight into which parameters should be given the most focus for improving understanding if more confident HCFC-141b emissions are desired. These simulations are driven by the prescribed error on each parameter, and thus, the results are highly dependent on both the estimated parameter values as well as their assigned uncertainties.






Figure 6 shows a similar temporal shape between our bottom-up emissions and those estimated from observations with a 12-box model (Western et al., 2022). The exception to this occurs at the very end of the time period when our calculation suggests a drop in emissions that seems inconsistent with the atmospheric observations. This supports the finding Western et al. (2022) arrived at using a hybrid bottom-up/top-down model. Barring production over the latest years that is substantially greater than reported, other changes in important parameters such as emission associated with production before sale, emissions associated with installation of foams, market shifts, etc. could lead to better agreement. However, any such changes would be larger than apparently required for the model to be consistent in temporal shape with observations at any other time during the history of HCFC-141b production and use. It is also unlikely that emissions from feedstock uses, which are not considered here, could explain the recent differences in emissions. Total production for intended feedstock use has been below 20 Gg/yr since the advent of feedstock use (Western et al., 2022) and it is expected that only a few percent of halocarbons used as feedstock will be emitted to the atmosphere (WMO, 2022). The relatively good performance of our model when compared with observationally derived emissions is a particularly informative result since it is a purely bottom-up method. That is, it does not adjust model parameters based on the observations.

While the shape of our modelled emissions matches the observationally derived emissions well, there is a consistent low offset in our calculations. It is unclear what is responsible for this. We have determined that elimination of the modelled phaseout of refrigeration uses over 2010-2015 does not improve the fit either in absolute magnitude or in the later trend. Higher emissions associated with production or installation or greater use of HCFC-141b as a solvent would shift the entire curve upward; however, the values required to bring the modelled center-line in agreement with the observations would be higher than what is generally accepted as likely. Despite the low bias, we suggest that the conclusion regarding the different emissions trends after 2017 remains valid.

In the calculation of the emissions shown in Figure 6, parameters in all regions have been assumed to be the same, aside from the decommissioning difference for the European/Japan region discussed in Section 2. Different parameters have been published for China (Wang et al., 2015), and different Weibull lifetime parameters for the Northeast Asian region (TEAP, 2019). If all Wang et al. (2015) parameters are adopted for the Northeast Asian region, there is a noticeable increase in global emissions later in the time series (Figure S6). While some product lifetimes are substantially shorter in Wang et al. (2015) relative to Table 1, those do not have a particularly large impact of global emissions calculated here. It is the very large emissions at the time of decommissioning for the Northeast Asian region that leads to most of the increase when comparing Figure S6 with Figure 6. Zhang et al. (2023) adopted most of the Wang et al. (2015) values, but did not use the decommissioning ones. If all the Wang et al. (2015) parameters are adopted for the Northeast Asian region except the decommissioning ones, and those are as in Table 2, there is little difference in global emissions (c.f., Figure S7). Similarly, if

the product lifetimes of TEAP (2019) are adopted for Northeast Asia, there is little change in global emissions (c.f., Figure S8). These calculation thus do not shed substantial information on why our emissions estimates are lower than those suggested by atmospheric measurements.

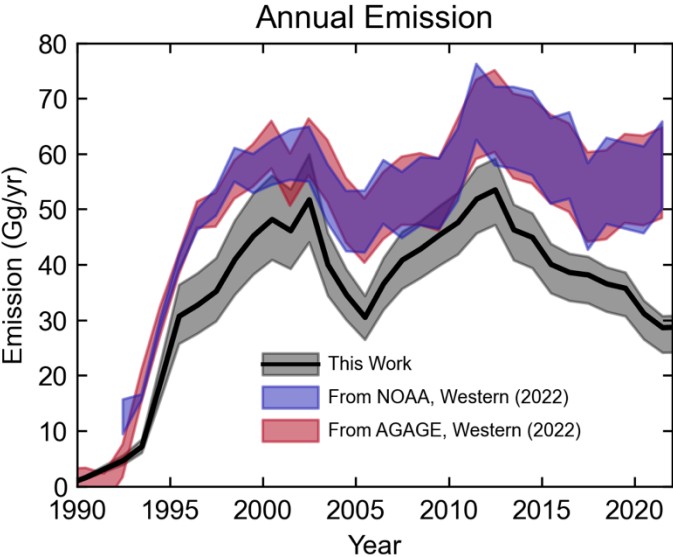

**Figure 6.** Total calculated global emissions compared with top-down estimates made with a 12-box model using NOAA and AGAGE atmospheric measurement networks, as given in Western et al. (2022). All shaded regions represent the 68% confidence range. Note that the overlap of the NOAA and AGAGE observations appears purple.

### 3.3 Global life cycle analysis

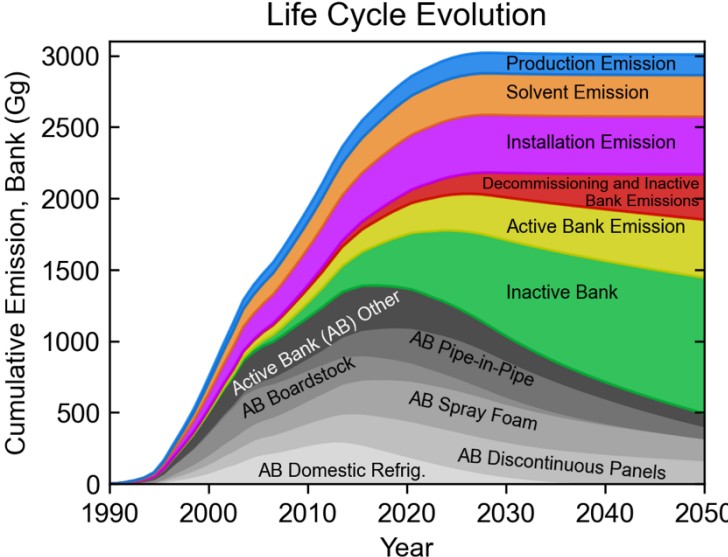

**Figure 7.** Life cycle analysis of all produced HCFC-141b over time. Emission quantities are cumulative over time and banks are instantaneous values. The top of the "production emission" curve represents total HCFC-141b that has been emitted or remains in banks. This is equivalent to cumulative consumption over time, including that which was not reported and was estimated here to be emitted as "Production Emissions", and excluding the small amount of HCFC-141b that is assumed to be captured and destroyed at the time of decommissioning in Europe. The second-to-the-top curve equals the cumulative reported consumption with the same decommissioning exclusionary caveat. The primary three active banks are identified, with the rest grouped together as "Other".

Figure 7 provides one approach to viewing the life-cycle analysis of HCFC-141b over time. It includes the sizes of the largest active banks and of total active and inactive banks, as well as the cumulative emissions from various emissions sources. By the middle of the century, the largest contributors to the active banks are pipe-in-pipe, spray foam, and discontinuous panels. By 2020, slightly more than half of all cumulative production to that point is calculated to reside in banks, with less than half having been emitted to the atmosphere. The amount emitted to the atmosphere continues to grow after 2020, coming entirely from the banks after production is assumed to cease from 2028 onward. We do not consider emissions from feedstock production or use in any of these calculations, which is currently believed to be very small (< 1 Gg/yr), as stated above, and is expected to continue after the phaseout of controlled production.

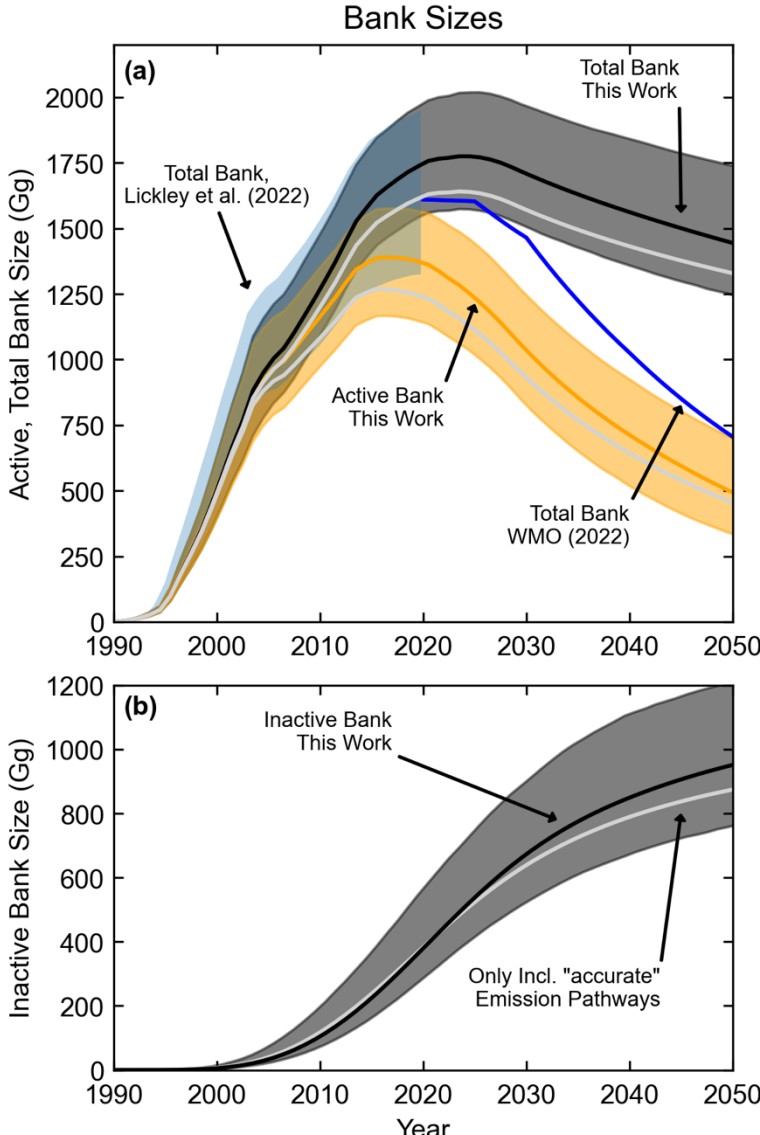

**Figure 8.** Calculated evolution of HCFC-141b banks. Panel a: The active bank for this work includes the total HCFC-141b found in all applications still in use in all regions and all markets. The total bank (active plus inactive banks) includes banks after decommissioning (i.e., landfills). Also shown are the total banks calculated by Lickley et al. (2022) for past years (blue shaded region) and by WMO (2022) for the future. The uncertainty range for this work is given as a 90% confidence interval, as is that for Lickley at al. (2022). Solid black and orange lines represent the banks for our baseline parameter values. Light grey lines that track in the lower half of the active, inactive, and total bank ranges represent the average bank sizes if only pathways are considered in which emissions fall within the emissions range estimated from observations (c.f., **Figure 6**) over most of the time period. Panel b: The inactive bank is separated out from its inclusion in Panel a to show its central value as well as its 90% confidence interval.

Figure 8 provides a more focused comparison of the bank estimates and shows the 2-sigma uncertainty range of our calculated

active, inactive, and total (active plus inactive) banks. The total bank compares well with that of Lickley et al. (2022). This is

despite the fact that the emissions calculated in this study are slightly lower than those estimated from observations, observations that were used as constraints in the Lickley et al. (2022) work. If we only consider sets of parameters from our

Monte Carlo analysis that lead to good agreement with the observationally-derived emissions, our bank estimates drop to the grey lines in the figure and remain in agreement with Lickley et al. given the size of the error bars. As previously mentioned, Lickley et al. (2022) found that an increase in reported production of 12% led to a better overall fit to emissions estimated from atmospheric concentration observations. In the set of our Monte-Carlo pathways that agree more closely with observationally derived emissions, the average emission associated with production is 8%. While this should not be considered

a retrieval of this value, it does show some level of consistency between the two studies in that they find better agreement with emissions estimated from observations when there are additional emissions relative to what is calculated from reported production. The comparison of our total bank with the bank projected in WMO (2022) is more complicated. While the starting values are similar, the WMO (2022) bank declines much faster than our total bank. Because the starting point for WMO (2022) was taken from Lickley et al. (2022), our agreement with Lickley et al. implies there must be good agreement with WMO

(2022) in 2020. After that, the different methodology here leads to differently shaped total bank curves. The WMO (2022) approach to calculate future emissions and banks assumes that the future bank release of HCFC-141b occurs at the same fractional rate of the total bank each year into the future. This approach does not allow for the fact that once in the inactive bank and after decommissioning emissions have occurred, HCFC-141b will almost certainly be released more slowly over time than when averaged over some portion of the previous life-cycle stages. Thus, it is expected that the total bank here would

decline more slowly than that of WMO (2022) after some amount of time. We can also compare our active bank estimates with those shown in Fig. 3.11 of TEAP (2023). When correcting for our low-emission bias as discussed above, while the active bank peaks around the same time, i.e., between 2010 and 2020, our estimate is about 40% higher. Also, it is clear that the active bank in our calculations drops off more slowly than those of TEAP (2023), with ours remaining close to 500 Gg in 2050, while theirs is close to 0. The reasons for these differences are unclear.

The separation of active and inactive banks, shown in Figure 8, has the potential to provide more useful information for policymakers regarding any potential climate and/or ozone benefit of mitigating bank emissions than projections of solely total bank values can. This is because capturing foams in large amounts that are already in landfills is particularly challenging. The ozone assessment projection for the total bank of HCFC-141b in 2030 is about 1400 Gg while ours is about 1750 Gg; however, our calculations show that almost 40% of our total bank will have been landfilled by that time and will no longer be

in products that are in use. The specific product/application that contains the foams further affects the feasibility of capturing the HCFC-141b; in fact, the amount of the 2030 bank in all refrigeration foam applications will only be about 50 Gg. We calculate that the primary active banks now and in the future will be in foams used in buildings, most of which have been found to be expensive, perhaps even prohibitively so, to recover (Caleb Management Services Ltd., 2010; ICF International, 2011).

A regional analysis of emissions and banks can be useful for understanding which regions are responsible for elevated atmospheric mole fractions and where opportunities might lie there were a desire to try to capture and destroy banks before

they are released. Figure 9 shows this information through 2040. North America and Europe dominated both emissions and active banks early in the time period, while Northeast Asia plays a much larger role later. By 2040, Northeast Asia's active bank is about 55% of the global active bank. While the active banks are still roughly half of their peak value by 2040, accessibility is likely much less than it would have been if refrigeration was the predominate contributor to the active bank (see Figure 7). It is important to remember that goods that are imported and exported, which already contain an ODS, are not reported as importing or exporting the ODS, itself. This could have implications for specific regions where emissions occur during the use phase and after. It could also impact exactly where the banks reside.

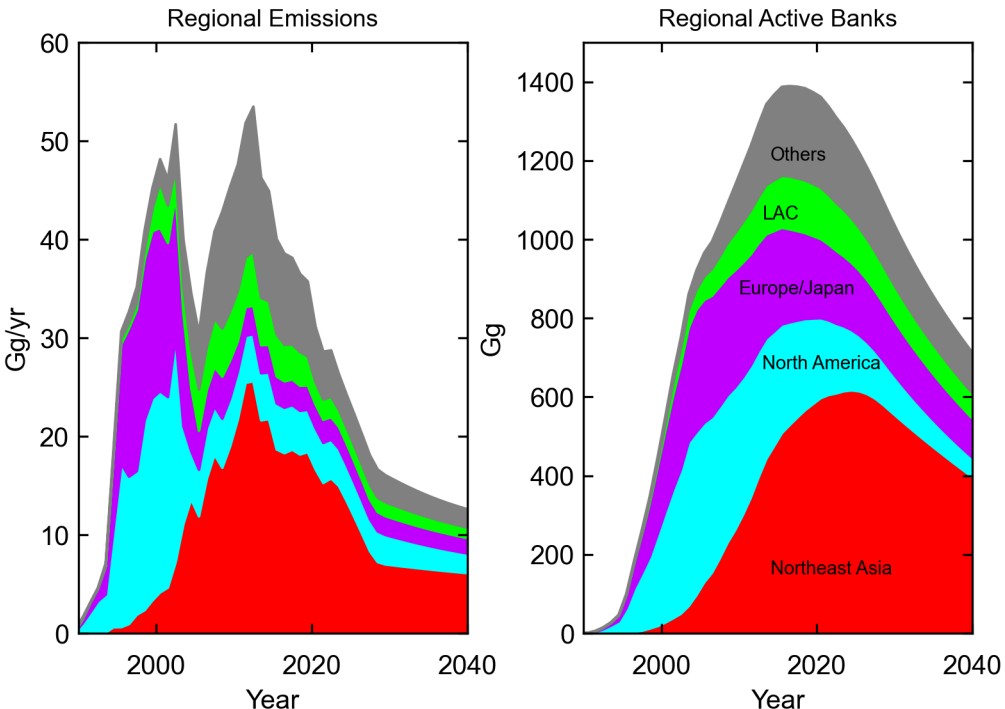

**Figure 9.** Stacked plot of regional contributions to HCFC-141b global emissions (left panel) and global active bank values (right panel). Color coding for the regions is in same in both panels, with the order of stacking determined by the cumulative emission through 2020, beginning with the region of largest emission at the bottom.

## 4 Conclusions

We have presented a new, bottom-up model that calculates the amounts of foam blowing agent residing in each life-cycle stage of the foam and the emissions that occur in each of these stages. The model incorporates reported production and published market information and emission factors. We have applied this model to HCFC-141b, which is a compound controlled under the Montreal Protocol. The calculations are performed for 10 geographic regions and for 11 foam markets. HCFC-141b was chosen for this work primarily because it is characterized by one of the most complete datasets of the controlled ozone-

depleting substances. Production was already required to be reported under the Montreal Protocol by the time HCFC-141b began to be used in substantial quantities, and atmospheric observations were also well established by that time. The model is not constrained to atmospheric observations, and thus represents an independent calculation of emissions and banks over time, unlike many other approaches that are constrained to observations in some manner (e.g., Lickley et al., 2022; Western et al., 2022; WMO, 2011, 2014, 2018, 2022; Velders and Daniel, 2014). Such a bottom-up approach can provide key information

regarding compliance with international agreements.

Our model provides information about banks in each specific application in which HCFC-141b has been used, thus allowing for a much better evaluation for the feasibility of capturing banks before they are released to the atmosphere. While the approach presented allows for including changing fundamental emission parameters over time, that has only been done here

for recovering banked HCFC-141b in refrigerator foams in Europe from 2002 onward, with sensitivity calculations made with varying emission parameters for the Northeast Asian region. Although other changes have likely occurred, there is not enough information for us to confidently make any other modelling adjustments. We have attempted to be liberal with our uncertainty estimates, however, to account for potential changes.

The comparison of calculated emissions with emissions estimated from global atmospheric measurement networks is quite good in terms of the temporal shape from the beginning of HCFC-141b use through the late 2010s, although our modelled emissions are generally somewhat lower. The most straightforward parameter change that would bring the calculated emissions higher across all years is to increase the emissions associated with production or to increase the fraction that has been used as a solvent. Importantly, however, these are not the only ways to improve agreement. For the last few years, our calculated

emissions and measurement-derived emissions increasingly diverge, suggesting there may be growing additional sources of emissions not included in the model, which could have relevance to the question of compliance with the Montreal Protocol. It could also be that model parameters changing over this time period may have caused some or all of the discrepancy; however, these changes have not been required to match the temporal shape of observationally derived emissions over the entire previous calculation period.


Historical total bank calculations compare well to those of Lickley (2022) within the 2-sigma error bars of both studies. Future total bank estimates begin in good agreement with those of WMO (2022) but the size of our bank estimates drops off more slowly over time. This difference is not surprising, given the differing approaches to projecting future banks. Future projections of active bank sizes calculated here are, of course, smaller than the total banks. This implies that the window of time is perhaps

somewhat shorter than one might expect from the results of Daniel and Reimann et al. (2022) if it is desired to intervene and keep the HCFC-141b in the banks from being released into the atmosphere at some point in the future. Furthermore, our analysis shows that by 2040, the majority of the HCFC-141b in banks will be found in spray foams and discontinuous panels, both used in buildings, and thus likely more expensive to extract before building demolition than capturing an ODS from, for

example, refrigeration units. Even in 2030, the more easily accessible foam banks residing in domestic and commercial
refrigeration and refrigerated containers (reefers) comprise less than 10% of the total active bank and less than 5% of the total bank.

The model presented could also be used to explore other ODSs and hydrofluorocarbon (HFC) banks and emissions from foam applications as well as from other non-foam applications, such as fire-fighting equipment and use as refrigerants. Furthermore,
here we have performed all calculations by starting with values for each of the input parameters along with their uncertainties. It would also be possible to use the model in combination with emissions estimated from atmospheric measurements to constrain some of the key parameters so the fit to the observations would be improved. Doing so would mean that the estimated emissions would no longer be independent of the observationally derived emissions and that calculated emissions would no longer be as clear of an indication of compliance with the Montreal Protocol. On the other hand, this approach would yield
bank estimates that are more consistent with the observationally derived emissions even if not necessarily more accurate by specific application. A challenge to overcome would be that with so many parameters, many of which lead to similar emission shape changes over time, the correlations affecting parameter retrieval would have to be explored in a careful way.

**Code availability.** The code used for this paper is publicly available and can be found at
https://csl.noaa.gov/groups/csl8/modeldata/.
**Data availability.** The data used for this paper can also be found at https://csl.noaa.gov/groups/csl8/modeldata/.
**Author contribution.** HWT developed the model on which the manuscript is based. JD wrote the Python software implementing the model. HWT and JD wrote the manuscript draft. CT designed the Fig. 1 and Fig. 2 schematic, and reviewed and edited the manuscript. LW provided emissions estimates from observations, and reviewed and edited the
manuscript.
**Competing interests.** The authors declare that they have no conflict of interest.
**Acknowledgements**. We thank members of the AGAGE and the NOAA measurement teams for their long-term monitoring of trace gases in the atmosphere and making those measurements publicly available. We thank the Ozone Secretariat for providing production, import, and export data that were used here. We thank Dr. David Fahey for helpful comments on a draft
version of this paper.

# Appendix A

Table A1. Sources used to determine parameters presented in Table 1. TEAP is the Technology and Economic Assessment Panel of the Montreal Protocol.

| Application | Emission During Manufacturing and Installation | Annual Emissions During Foam Use | Failure Rate Parameters for Weibull function | |
| --- | --- | --- | --- | --- |
| | | | Weibull Scale Factor | Weibull Lifetime Term |
| **Refrigeration** | | | | |
| Domestic refrigeration | TEAP assessment based on foam installation practices and what is given in the "FTOC" column in Table A4.3 of TEAP (2019). Values are greater than or equal to measurements for panel installation for domestic refrigerators (Aprahamian and Bowman, 2005). All values are identical to the referenced table except for discontinuous panels, which is prescribed to be 10% rather than 20%. | Same as Table A4.3 of TEAP (2019). Values are greater than or equal to the change in conductivity measured in the Wilkes et al. studies (Wilkes et al., 2001; Wilkes et al., 2003) | (UNFCCC, 2017) | Table 1 of UNFCCC (2017) |
| Commercial refrigeration | | | | DOE (2014), TEAP (2019) |
| Refrigerated containers | | | | From EPA (2011), with value between land- and ship-based containers |
| **Building Construction** | | | | |
| Discontinuous Panels Continuous panels Spray foam | | | From global residential results in Deetman et al. (2020); other relevant references are Aktas and Bilec (2012) and Andersen and Negendahl (2023) | From global residential results in Deetman et al. (2020); other relevant references are Aktas and Bilec (2012) and Andersen and Negendahl (2023) |
| PU boardstock | | | Used in TEAP (2019), but not in tables | FTOC column of Table A4.3 in TEAP (2019) |
| **Other Uses** | | | | |
| PU block and pipe | | TEAP (2019), assuming value in table A4.3 is a factor of 10 too large (see Sect. 2.3 text); varying values from 0.0075 to 0.75 | Used in TEAP (2019), but not in tables | FTOC column of Table A4.3 in TEAP (2019) |

| | | | | |
|---|---|---|---|---|
| | | matters little to results | | |
| PU pipe in pipe | | Treated like discontinuous and continuous panels in *TEAP* (2019) due to unexpectedly large value in TEAP table (also see Sect. 2.3 text) | | 15 yrs was used in the calculations of TEAP (2019) owing to unpublished suggestions that previous lifetime assumptions were too large. Table A4.3 provides values of 50 and 75 yrs., so here, we compromise and use 30 yrs. |
| PU block foam slab | | (TEAP, 2019) | | (TEAP, 2019) |
| PU integral skin | Taken as smaller than many references to account for the nuance that some integral skin manufacturing results in open cells and some results in closed cells; however, due to small market for HCFC-141b, our assumption is rather insignificant. | Assumed based on skin sealing the cells and allowing for only slow release | Assumed to be the same as "Other Uses" in Table 1 of this work | Used in TEAP (2021), but not shown |


**Table A2.** Sources used to determine parameters presented in Table 2.

| Type of Emission | Magnitude and Timing |
|---|---|
| Before sale (includes loss during production) | Consistent with TEAP (2019), although arguably on the lower end when considering both advanced and less sophisticated production plants |
| During Decommissioning | Drawn from publications that have evaluated impact of foam shredding at time of decommissioning (Scheutz and Kjeldsen, 2002; Scheutz et al., 2007; Kjeldsen and Scheutz, 2003) and recently estimated ranges (TEAP, 2019) |
| From inactive bank (i.e., post-decommissioned bank such as landfill) | Used lower value of 0.5% from Table A4.3 (TEAP, 2019) for all applications |
| Solvent Use | See text of Sect. 2.1 |


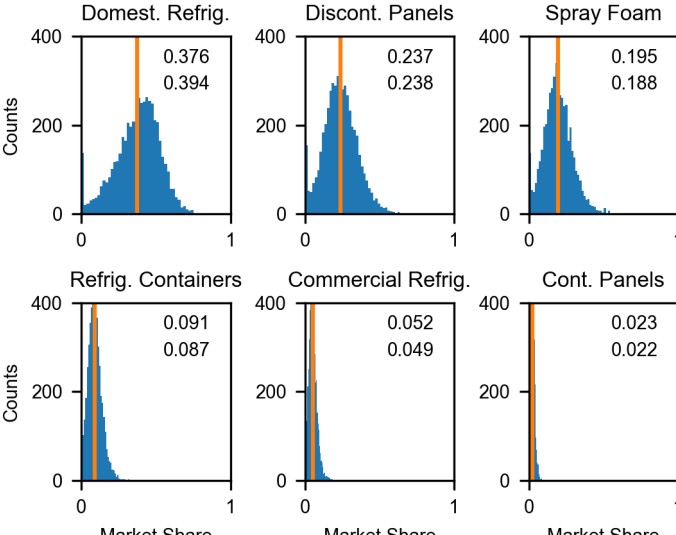

Figure S1. Probability distribution functions of market share for the 6 largest markets in 2008 for the Latin America and the Caribbean region. Abscissas represent fraction of market share with ordinates providing a histogrammed distribution with 5000 total cases run. The numbers in the boxes represent the prescribed mean of the distribution (lower) and the mean the model calculates after taking the normalization approach described in Section 2 (upper). The vertical orange line represents the mean of the distribution shown.

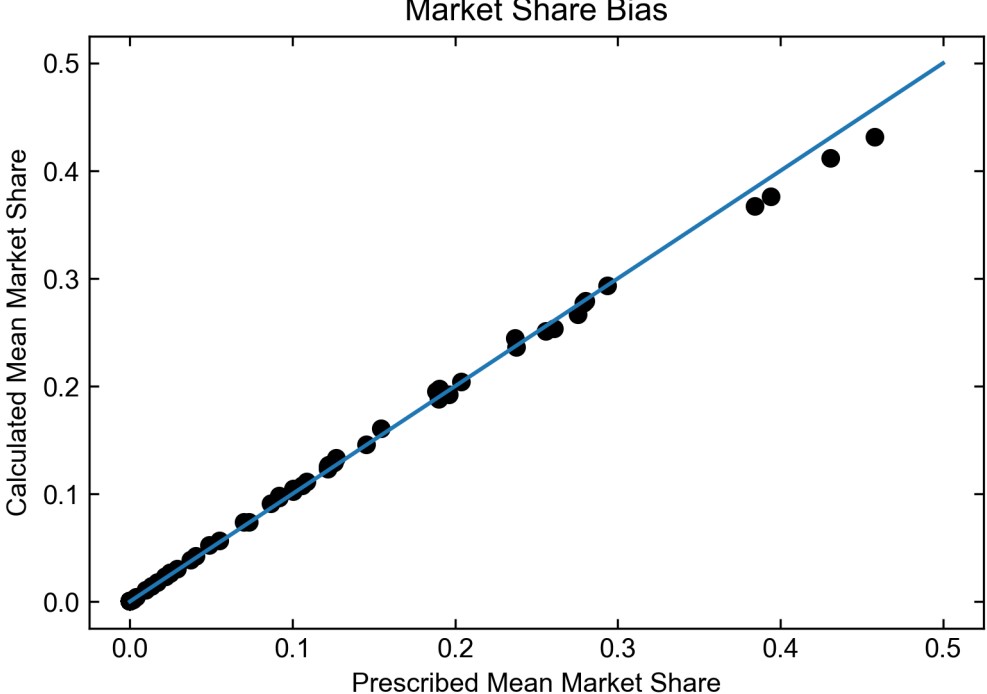


Figure S2. A comparison of the prescribed market share with what the model calculates using the Monte Carlo model discussed in Section 2. The deviation from the one-to-one line shown represents the bias inherent in this approach. These points are for the 2008 market share values across all regions before refrigeration uses are phased out.

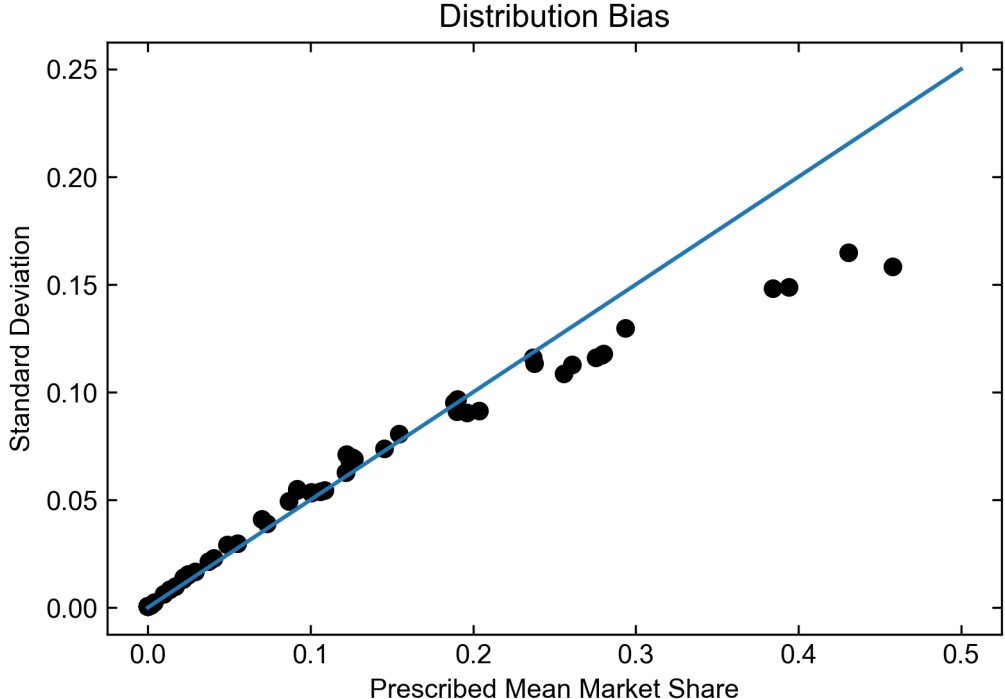

Figure S3. The calculated market share standard deviations as calculated by the model versus the prescribed mean market

share values for the same case as shown in Figure S2. The prescribed standard deviation is 50% of the market share value,

with the deviation from the line shown representative of the bias in this approach.

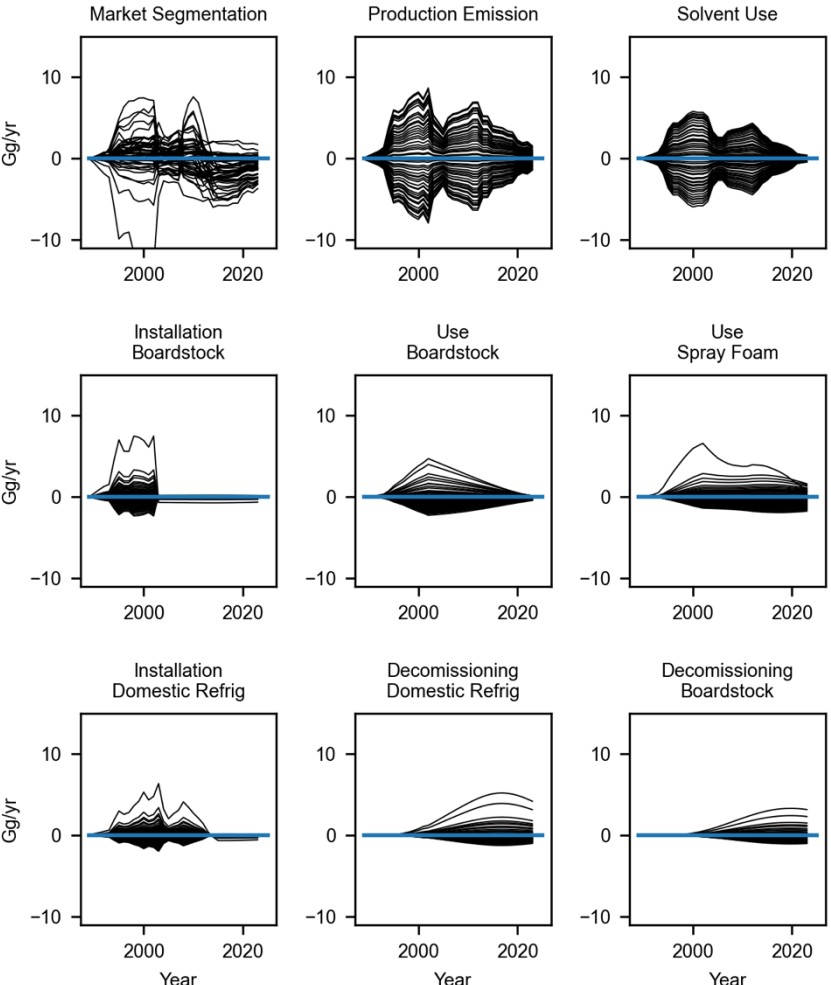

**Emissions Variability from Individual Input Paramters**

Figure S4. Response of calculated emissions to 50 Monte Carlo simulations in which input parameters were varied one at a time. The 9 parameters shown here lead to the greatest emissions changes in terms of mean standard deviation of time series over 1989 through 2024. Each time series represents the emissions time series calculated with the baseline input parameter (i.e., values from Tables 1 and 2) subtracted from the time series with the varied time series input parameter; each varied parameter is determined by its prescribed probability distribution function.


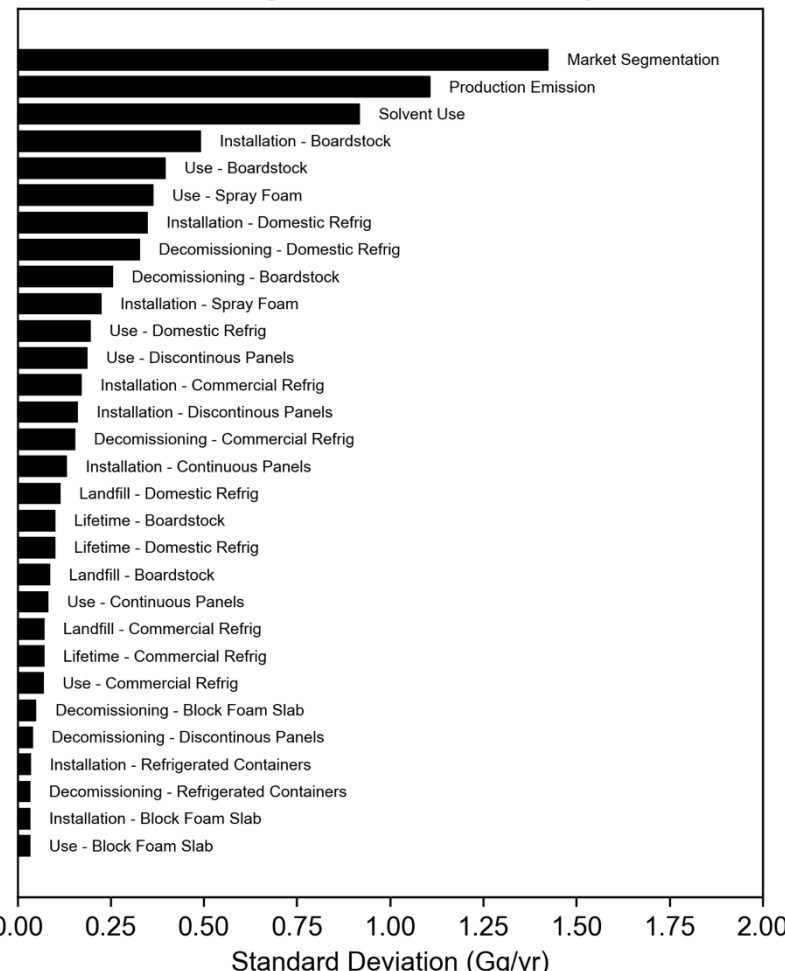

Figure S5. The average standard deviation of emissions for time series as calculated in Figure S4 over 1989 through 2024 due to changes in each input parameter; results are arranged from most significant to least for the 30 most significant parameters. Values are calculated from 50 Monte Carlo simulations.


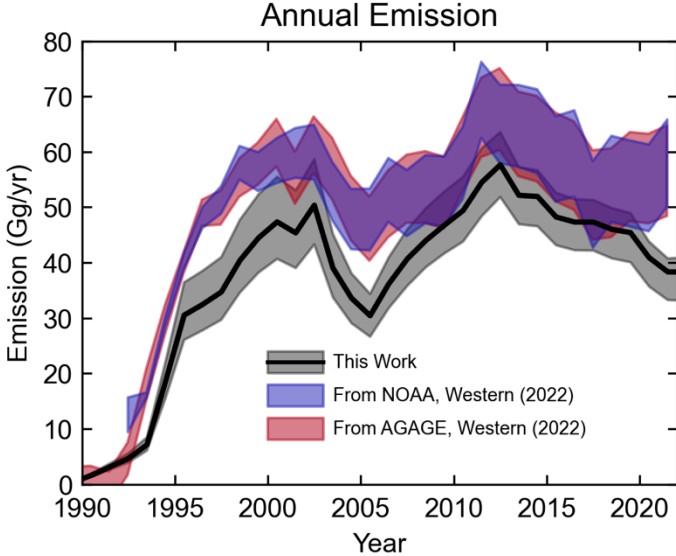

Figure S6. Global emissions calculated as for Figure 6, but with values for the Northeast Asian region taken from Wang et al. (2015). "Refrigeration" values are applied to our domestic and commercial refrigeration as well as refrigerated containers. "Pipeline" values are applied to our pipe-in-pipe values, and "sheet" values are applied to our discontinuous and continuous panel markets. All other values are the same as in Tables 1 and 2. We do allow emissions to continue after decommissioning at the rate we specified in Table 2. Results are calculated from 500 Monte Carlo runs.

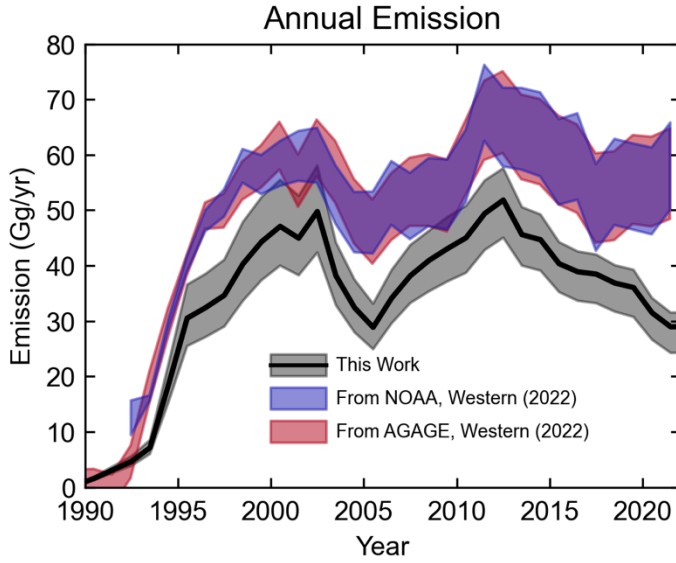

Figure S7. Same as Figure S1 except the decommissioning emission factors for the Northeast Asian region are taken from Table 2 rather than from Wang et al. (2015).

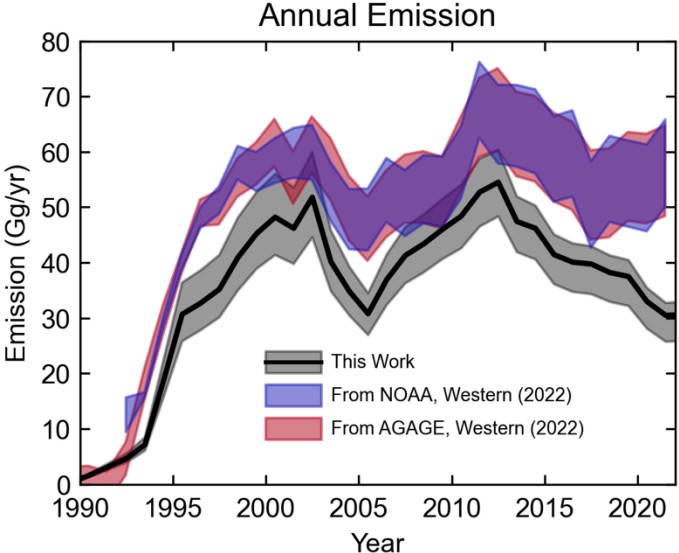

Figure S8. Global emissions calculated as for Figure 6, but with lifetime values for the Northeast Asian region taken from TEAP (2019). All other values are the same as in Tables 1 and 2.

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
