# Peer review of "A new production-based model for estimating emissions and banks of ODSs: Application to HCFC-141b"

_EGUsphere, 2025_

## Author Response (AR1)

**Guus Velders Reviewer**

**We thank Dr. Velders for the time he spent on our manuscript. We found his comments very helpful and have tried to address them all below. In fact, we are fortunate to have insightful comments from three reviewers, each of whom clearly spent quality time with our manuscript.**

**All line numbers refer to the previously submitted pdf. We have put all our response information in bold with the reviewer comments in plain text.**

The authors discuss novel work on deriving emissions with a bottom-up approach taken into account all sectors in which HCFC-141b is being used. Although HCFC-141b is a minor ODS, the method that is developed is very useful to be applied to other ODS and F-gases in general. Especially the estimates of active and inactive banks with a potential for mitigation options makes the work relevant for policymakers.

The paper is  well written and the methods well described.

**Thank you.**

The title refers to a new model and the abstract says that a refined model has been developed. However, the abstract does not say anything about this model and what is new about it. It does mention important results derived from the model. So, what is the focus of the paper, the new model or the results? I suggest you clarify this and at least write something in the abstract what new is in the model.

**To clarify what is new in the model, we state that we present a new model:**

**"… that incorporates existing use and life-cycle information to calculate emissions and banks as well as uncertainties in the quantities."**

**We also separate the sentence discussing the model and the application it is applied to in order to add more clarity that it is primarily the model development that is the focus of this study. While the application to HCFC-141b yields some very interesting results, the model can be applied much more widely. The new sentence now starts:**

**"To demonstrate the model, we apply it to 1,1-dichloro-1-fluoroethane (HCFC-141b), …"**

**We take the same splitting approach in the last paragraph of the introduction.**

**We have also altered the beginning of the conclusions to make it clearer that the model is the primary result, with the application to HCFC-141b secondary.  We now state:**

**"We have presented a new, bottom-up model that calculates the amounts of foam blowing agent residing in each life-cycle stage of the foam and the emissions that occur in each of these stages. The model incorporates reported production and published market information and emission factors. We have applied this model to HCFC-141b, which is a compound controlled under the Montreal Protocol."**

What I miss are results for the different geographical regions. It is mentioned that the method is applied to 10 regions, but no results are given. Information on where active and inactive banks are located would be important for policymakers and for the potential of mitigation options.

**We have now added a figure (Figure 9) to the results section showing regional contributions to global emissions and active bank values. This is accompanied by a new paragraph of discussion.**

Related to lines 358-360 and Figure 5: It is mentioned that the emissions from your work are similar that the emissions derived from the NOAA and AGAGE networks, but that there is a discrepancy in the last few years. But there is an absolute difference in emissions of 10-20 Gg/yr. If you take this into account the discrepancy after about 2017 is less clear.

**We have added text to Section 3.2 to address this offset:**

**"While the shape of our modelled emissions matches the observationally derived emissions well, there is a consistent low offset in our calculations. It is unclear what is responsible for this. We have determined that elimination of the modelled phaseout of refrigeration uses over 2010-2015 does not improve the fit either in absolute magnitude or in the later trend. Higher emissions associated with production or installation or greater use of HCFC-141b as a solvent would shift the entire curve upward; however, the values required to bring the modelled center-line in agreement with the observations would be higher than what is generally accepted as likely. Despite the low bias, we suggest that the conclusion regarding the different emissions trends after 2017 remains valid."**

Also, how much are the emissions in the latter years affected by your assumption given in L207-209 that HCFC-141b in refrigeration is linearly phased out over 2010-2015?

**If we do not allow refrigeration uses to phase out, the largest emissions difference in any year is less than 2 Gg and doesn't alter any of the conclusions presented. We now add the following text:**

**"If we do not apply this phaseout, the largest annual difference in calculated emission is less than 2 Gg, and it does not affect the discussion in Section 3."**

I can also not find how the market splits was after 2015. What is assumed for these latter years and how much does that effect the trend in emissions past 2015.

**We have clarified the assumptions (new text enclosed in brackets for clarity here) for the 2015 and onward time period in Section 2.2 with the text:**

**"Values from 2008 [through the end of the calculated time period] are given by UNEP (2010),…**

Some specifics comments:

L78-80, : This is probably true for banks, but not for top-down derived emissions. The uncertainties in top-down inferred emissions are generally much smaller than from bottom-up derived emissions.

**We would like to retain the part about the biases since it is accurate (i.e., even for emissions, if the lifetime is wrong, the direct estimates of emissions from observations will be biased), and yet do indeed agree with the point that top-down emissions estimates are generally thought to be more accurate.**

**We therefore now make lines 94-95 (new text in brackets here):**

**"However, associated with the model flexibility and wide range of model inputs are important data gaps that can result in [banks and emissions values characterized by large uncertainties and potential biases, which are generally thought to be larger than those associated with top-down emissions estimates.]"**

L105-106: I suggest you give the reference for the production data here. Also, refer to, e.g., WMO2022, for a reference for the observations of mole fractions.

**We have added these references.**

L139: Great graphical representation of the different stages and cumulative emissions.

**Thank you.**

l145-146: Countries report data of the individual HCFCs to UNEP, but only the aggregated data for total ODP-weighted HCFCs is published by UNEP. I assume you used the data from the individual HCFCs and did not disaggregate the ODP-weighted total HCFCs data. That is probably why the data is summed per region. I know, referencing the real data you used is than tricky (just a remark, no solution).

**This is a good point.  We have added a new sentence after this one pointing out that we used data reported specifically for HCFC-141b that was provided to us by the Secretariat:**

**"While compounds are reported as being aggregated by compound groups, the Secretariat has provided specific HCFC-141b values to us with the agreement that no data or results will be shown for any specific country."**

**We have also added the Secretariat to the acknowledgments.**

L218: I suggest you give the value of the low boiling point here, to support the statement that emissions will easily occur also in more or less confined applications.

**We now add the boiling point, parenthetically.**

L222-223: What is the reason you let the emissions decrease for large installations?

**We have clarified this sentence since it was confusing before. This sentence now has become:**

**"We assume an absolute uncertainty standard deviation of 5% on all emissions associated with installation. For example, if the installation emission fraction is 15%, its uncertainty range is 15%±5%."**

L223: "larger estimated installation emissions". What are installation emissions? Should this be equipment, production or use?

**As in response to the previous comment, we have tried to clarify this previously confusing sentence.**

L256: "assuming the quoted value is a factor of 10 too large". You can not just write "is taken from Table A4.3 in TEAP (2019)"and then divide the value by a factor of 10. Please justify this.

**The reviewer is absolutely correct. We now provide more explanation of this value and point out that the particular choice for this work makes very little difference because block-pipe is a relatively small market for HCFC-141b. We now state:**

**"Our block-and-pipe emission rate estimate is taken from what was used in TEAP, 7.5% (TEAP, 2019), although Table A4.3 in TEAP (2019) mistakenly stated that 75% was the value used. To further complicate matters, some references have used 0.75% for HFC emissions (IPCC/TEAP, 2005; IPCC, 2006). For this work, because the block-and-pipe market is relatively small for HCFC-141b, what we use for this value is of very small relevance. The largest annual difference in emissions between using 0.75% and 75% is less than 0.7 Gg. "**

L225, Table 1: For the Weibull function you refer to section 2.4. Shouldn't that be 2.3?

**It should really be both since the functions are described in Section 2.3 and decommissioning is discussed in 2.4. Thank you. This has been changed.**

L325: Section 2.6: In the introduction you mention regional differences in emissions, from which I assumed that this would be taken into account in the model. Is this the case or not? In the conclusion you mention again that the analysis is performed for 10 geographical regions, but now data or figure with emissions of banks is presented for the regions? Please be specific how the regions are taken into account in the modelling.

**In response to this very helpful comment and a comment from another reviewer, we have added regional contributions to global emissions and the active bank. We have also added a paragraph of discussion for this figure in the results section. It states:**

**"A regional analysis of emissions and banks can be useful for understanding which regions are responsible for elevated atmospheric mole fractions and where opportunities might lie there were a desire to try to capture and destroy banks before they are released.** Error! Reference source not found. **shows this information through 2040. North America and Europe dominated both emissions and active banks early in the time period, while Northeast Asia plays a much larger role later. By 2040, Northeast Asia's active bank is about 55% of the global active bank. While the active banks are still roughly half of their peak value by 2040, accessibility is likely much less than it would have been if refrigeration was the predominate contributor to the active bank (see** Error! Reference source not found.**). It is important to remember that goods that are imported and exported, which already contain an ODS, are not reported as importing or exporting the ODS, itself. This could have implications for specific regions where emissions occur during the use phase and after. It could also impact exactly where the banks reside."**

L292: Figure 3: I suggest you make clear that what is shown is not the total bank, but the active bank (see text above the figure).

**This is a good point. We have done this both in the text and in the title to the figure.**

L326-328: I suggest you refer here to Velders and Daniel (2014) who performed a similar Monte Carlo analysis.

**We have added this reference.**

L343-345: How the text now reads, it seems that the market breakdown is completely new in this paper, while from section 2.2 it is clear that is based on various UNEP/FTOC reports (with some additional assumptions). Please mention this here

**We have added information showing the information is not completely new here. The sentence now reads (new text inside brackets):**

**The market segmentation approach described in Sect. 2.2, [with information taken from UNEP (2003a), UNEP (2007), and UNEP (2010)], yields a market breakdown over time shown in** Error! Reference source not found.

**We have also added additional text in the abstract since we feel this point is so important. This text was stated above and is (relevant new part in brackets):**

**"Here, we present a new bottom-up model [that incorporates existing use and life-cycle information] to calculate emissions and banks as well as uncertainties in the quantities."**

**RC1 Reviewer**

**We thank this reviewer for the time they spent on our manuscript. We found their comments very helpful and have tried to address them all below. In fact, we are fortunate to have insightful comments from three reviewers, each of whom clearly spent quality time with our manuscript.**

**All line numbers refer to the previously submitted pdf. We have put all our response information in bold with the reviewer comments in plain text.**

The study by Walter-Terrinoni et al. intends to present a new bottom-up model for HCFC-141b, a chemical used primarily in foam insulation and whose production is currently being phased out. Using this model, global emissions for HCFC-141b are calculated and compared to measurements. For the time period 1990-2017 the authors find a good agreement with the measurements, but after 2017 the model underestimates the emissions. The authors explain this discrepancy between measurements and model by either a growing additional source of emissions that is inconsistent with reported production or a model deficiency that did not exist or was not apparent before 2017. The manuscript is generally well written and deserves to be published, but major revisions are needed before manuscript can be accepted for publication.

**General comment:**

In the title, abstract and conclusion it is stated that a new model is presented. However, I could either clearly understand how your model works since I could not find a model description in the manuscript or what the new in this model. Thus, the title, abstract and conclusion do not fit to the content presented in the manuscript and need to be adjusted and the method sections needs to be rewritten so that a model description is provided.

**We think we understand the source of confusion here. The model actually calculates emissions, rather than using them as input to calculate impacts of the emissions. We**

**have added some clarification to line 14 of the abstract and lines 111-113 of the introduction. The abstract sentence now is:**

**"Here, we present a new bottom-up model that incorporates existing use and life-cycle information to calculate emissions and banks as well as uncertainties in the quantities."**

**The sentence starting on line 111 is now:**

**"Here, we present a bottom-up model that calculates banks and emissions from foam applications."**

**Perhaps more important, we have added an equation to Section 2 for the total emissions that we calculate. We hope that this approach makes it more clear how each part of Section 2 describes a different part of the model calculation. We also explicitly include equations for these terms in the subsections of Sect. 2.**

**Specific comments:**

P4, L119ff: The method section is quite lengthy and the emission values used are explained too much in detail. I had the feeling that I rather read here a scientific report for policy makers than a scientific paper.

**As we replied to the general comment, this primary purpose of this model is actually to calculate emission values from all the inputs discussed in the Methods section. Hopefully the previous response will fix this source of confusion and make it clear that emission values are the primary output of the model.**

P6, L155-L157: For this statements a reference is missing. Where is this documented? Or is this a result from your study? Or are you referring here to some figure shown in the manuscript?

**This is a good point. We have tried to clarify why this is true by adding some detail that was missing. Beginning with the second sentence of this paragraph, it now states:**

**"In the global total, imports should be equal to exports. Thus, if reporting were perfect, by definition consumption and production would be nearly equal in the global sums at least when summing over several years. Even with perfect reporting, individual years might have slightly unequal consumption and production if some quantities were to be attributed to the next year due to supply chain delays between the timing of export and import."**

**P6, L159 and L169: What are non-Article 5 countries and what are non A5 countries. Same holds for A5 countries, which exactly belong to A5? Where does this naming come from?**

**We have added a new map (Figure 2) that shows which countries are Article 5 under the Montreal Protocol and which are non-Article 5. We thank the reviewer for catching our use of this jargon. We have also show on this map the various regions that we consider in our calculations.**

P6; L162: This statement is not in accordance with Fig. 2. The highest emissions are for both peaks for the yellow stack which is labelled with "others".

**We have added the clarification to the first sentence of the caption that this is a "Stacked plot of HCFC-141b consumption…". So the consumption for "Others" is only the part shaded in yellow; it is not the total from 0. We have also added a new sentence to the figure caption:**

**"The top of the "Other" region represents the global total consumption."**

**Additionally, we have added a line at the very top of the curves and labeled it "Global Total".**

Figure 2: Why have the "other" countries the highest emissions? Shouldn't that be rather one of the industrial countries?

**Please see previous response.**

P11, L286-290: Where is the model description and what is new? Are the five equations given here describe the model?

**We have rewritten the beginning of the methods section to hopefully make this clearer. It starts:**

**"The life-cycle stages during which HCFC-141b emissions occur and are calculated by our model are shown in Fig. 1. The rest of this section will describe how our model calculates these emissions for different applications. It is the sum of the emissions over the entire life cycle of each market that represents total emissions at any given time. For each market**

$$E_i^{total} = E_i^{production} + E_i^{solvent} + E_i^{install} + E_i^{use} + E_i^{decom} + E_i^{landfill} \qquad (1)$$

**where $E_i^{total}$ is the total emission of HCFC-141b in year *i*. In the equations that follow, we assume there is only a single market, for simplicity."**

**We hope that this, combined with changes made to the previous responses of this review help clarify what is calculated."**

P13, L326: Here you refer to the above described model description which I as reader could not find. For me the previous section was a summary of emission assumptions that have been

used to run the model, thus rather which input values have been used rather than how the actual calculation has been done.

**We hope the previous alterations now make it clear that Section 2 shows how emissions are calculated in each life cycle stage.**

P14, L340: Reaching the end of the method section still leaves me puzzled with the questions on what is new and how does the model work. For me this did not become clear only which assumptions have been made.

**We hope the previous alterations now make it clear that Section 2 shows how emissions are calculated in each life cycle stage.**

P15, L358: Add which observations have been used.

**Thank you for catching this. We have now added a reference for the observations.**

Figure 5: Looking at this figure I have two questions: (1) Your model is generally underestimating the measured emissions. Do you have any idea why? Do you have an idea what could be missing in your model or is this due to an inaccuracy or bias from the measurements? (2) Is this a comparison with pure observations or with models that use observations?

**This is a good question, which we did not attempt to address before because it seemed our thoughts might be considered too speculative. However, the reviewer is asking a question that would be likely asked by most readers. We have now added a paragraph of discussion in relation to the discussion of the emissions figure. The bottom line is that we cannot know the reason for the low bias from this study. The new part of the discussion is:**

**"While the shape of our modelled emissions matches the observationally derived emissions well, there is a consistent low offset in our calculations. It is unclear what is responsible for this. We have determined that elimination of the modelled phaseout of refrigeration uses over 2010-2015 does not improve the fit either in absolute magnitude or in the later trend. Higher emissions associated with production or installation or greater use of HCFC-141b as a solvent would shift the entire curve upward; however, the values required to bring the modelled center-line in agreement with the observations would be higher than what is generally accepted as likely. Despite the low bias, we suggest that the conclusion regarding the different emissions trends after 2017 remains valid."**

**For the second part of this question, the second paragraph of section 3.2 now starts:**

**"Figure 6 shows a similar temporal shape between our bottom-up emissions and those estimated from observations with a 12-box model (Western et al., 2022)."**

P18, L424: Here you partly answer my first question in the previous comment. It seems that you cannot explain this differences, but do you have any idea?

**See response to previous comment.**

P19, L444: I would rather name this section "Discussion and conclusion".

**Because of the discussion that is currently in Section 3, it seems that "Conclusions" is more appropriate here. However, we are happy to change this if the editor agrees.**

P19, L446: Which "10" geographic regions? In which regions you have separated into has nowhere been mentioned.

**We really thank the reviewer for this comment. This and the comment about article 5 countries have motivated us to add a new map with this information on it (Figure 2).**

P19, L446: Also a list of the 11 foam markets should be added somewhere in the manuscript.

**In the first paragraph of Section 2.2, we have added a reference to Table 1, which lists the 11 markets we consider. The relevant sentence is:**

**"To obtain information about the division of HCFC-141b between the various markets most important for HCFC-141b use and considered here (see Table 1 for list of markets)…"**

P20, L493: A clear statement what the implication of your study are is missing.

**We hope that with the additional clarification that the model, itself, is the primary addition here that the conclusions of the study now make more sense. Even so, the results of the application to HCFC-141b are very interesting, and may even have relevance to compliance with the Montreal Protocol, as we state in the conclusion. We have made this last point more directly by adding to the third paragraph of the conclusions, where we discuss the global emissions comparisons:**

**"… which could have relevance to the question of compliance with the Montreal Protocol."**

**Technical corrections:**

P11, L267: remove parentheses around the references.

**Done.**

P15, L358: Figure 5 show -> Figure 5 shows

**Done**

P19, L439: emissions time series -> emission time series

**Done**

P19, L442: I guess you mean here rather "noting" than "nothing".

**Yes.  Thank you.**

P19, L452:  remove parentheses around the references.

**We thank the reviewer for the technical corrections.  We have done these.**

**RC2 Reviewer**

**We thank this reviewer for the time they spent on our manuscript. We found their comments very helpful and have tried to address them all below. In fact, we are fortunate to have insightful comments from three reviewers, each of whom clearly spent quality time with our manuscript.**

**All line numbers refer to the previously submitted pdf. We have put all our response information in bold with the reviewer comments in plain text.**

**General comments**

This manuscript presents a bottom-up model to estimate global emissions and banks of HCFC-141b based on a comprehensive product lifecycle framework. The study incorporates a wide range of sectoral uses and applies Monte Carlo uncertainty analysis to identify key drivers. The methods are generally robust, and comparisons with atmospheric observations show good temporal agreement. The paper provides valuable insights into bank composition and mitigation potential, which are relevant for policymakers.

However, several assumptions—particularly in parameter selection and uncertainty treatment—warrant further justification. Additionally, the structure of the abstract and limited discussion of regional outputs slightly weakens the policy relevance of the results. Clarifications on model improvements and regional trends would enhance the manuscript's clarity and utility.

**As we have stated in response to comments by Dr. Velders, we have tried to elevate the emphasis to the model itself through some text changes in the abstract and in the introduction. Also, as discussed in response to a comment below, we have added a figure in the main text that shows the regional contributions to emissions and active banks. This is a repeat of what we responded to him:**

We also separate the sentence discussing the model and the application it is applied to in order to add more clarity that it is primarily the model development that is the focus of this study. While the application to HCFC-141b yields some very interesting results, the model can be applied much more widely. The new sentence now starts:

"To demonstrate the model, we apply it to 1,1-dichloro-1-fluoroethane (HCFC-141b), …"

We take the same splitting approach in the last paragraph of the introduction.

We have also altered the beginning of the conclusions to make it clearer that the model is the primary result, with the application to HCFC-141b secondary. We now state:

"We have presented a new, bottom-up model that calculates the amounts of foam blowing agent residing in each life-cycle stage of the foam and the emissions that occur in each of these stages. The model incorporates reported production and published market information and emission factors. We have applied this model to HCFC-141b, which is a compound controlled under the Montreal Protocol."

**Specific comments**

- Line128-135: The decision to assume uniform emission and lifecycle parameters across all regions (except Europe) overlooks the availability of region-specific data, such as those for China (Wang et al., 2015). The authors should discuss the implications of this simplification on regional accuracy or perform a sensitivity test incorporating these regional variations.

Thank you for this observation. We agree that we should address the Wang et al. data as well as some regional TEAP (2019) data. At first, we were going to apply their data to our calculations in the main text. However, after giving it more thought, we prefer to add these results to the supplement. The reason is because we have concerns with their large emissions at the time of decommissioning. This leads to substantially more emissions later in the time period, as seen in the new supplemental figure. To further complicate matters, Zhang et al. (2015) did not use these decommissioning values although they adopted the other Wang parameters. So we now also include a figure in the supplement with the Wang values, but with our decommissioning emission factors.

We added a new paragraph of discussion in the results section about this. We state:

"In the calculation of the emissions shown in Error! Reference source not found.**, parameters in all regions have been assumed to be the same, aside from the decommissioning difference for the European/Japan region discussed in Section 2. Different parameters have been published for China (Wang et al., 2015), and different Weibull lifetime parameters for the Northeast Asian region (TEAP, 2019). If all Wang et al. (2015)**

**parameters are adopted for the Northeast Asian region, there is a noticeable increase in global emissions later in the time series (Figure S6). While some product lifetimes are substantially shorter in Wang et al. (2015) relative to Table 1, those do not have a particularly large impact of global emissions calculated here. It is the very large emissions at the time of decommissioning for the Northeast Asian region that leads to most of the increase when comparing Figure S6 with** Error! Reference source not found.**.. Zhang et al. (2023) adopted most of the Wang et al. (2015) values, but did not use the decommissioning ones. If all the Wang et al. (2015) parameters are adopted for the Northeast Asian region except the decommissioning ones, and those are as in Table 2, there is little difference in global emissions (c.f., Figure S7). Similarly, if the product lifetimes of TEAP (2019) are adopted for Northeast Asia, there is little change in global emissions (c.f., Figure S8). These calculation thus do not shed substantial information on why our emissions estimates are lower than those suggested by atmospheric measurements."**

- Line 132-133: Since regional differences in consumption and market size significantly affect emissions and banks—ultimately influencing global estimates—I recommend that the authors include regional emission results in the main text to better support regional policy development.

**We have now added a figure showing the regional contributions to the total emissions and the active bank. We have also added the following discussion:**

**"A regional analysis of emissions and banks can be useful for understanding which regions are responsible for elevated atmospheric mole fractions and where opportunities might lie there were a desire to try to capture and destroy banks before they are released.** Error! Reference source not found. **shows this information through 2040. North America and Europe dominated both emissions and active banks early in the time period, while Northeast Asia plays a much larger role later. By 2040, Northeast Asia's active bank is about 55% of the global active bank. While the active banks are still roughly half of their peak value by 2040, accessibility is likely much less than it would have been if refrigeration was the predominate contributor to the active bank (see** Error! Reference source not found.**). It is important to remember that goods that are imported and exported, which already contain an ODS, are not reported as importing or exporting the ODS, itself. This could have implications for specific regions where emissions occur during the use phase and after. It could also impact exactly where the banks reside."**

- Line 255-257: Given that a change from 75% (referred in the 2019 TEAP report) to 7.5% is quite large, please provide the exact reason for choosing a scaling factor of 10.

**The reviewer is absolutely correct. We now provide more explanation of this value and point out that the particular choice for this work makes very little difference because block-pipe is a relatively small market for HCFC-141b. We now state:**

**"Our block-and-pipe emission rate estimate is taken from what was used in TEAP, 7.5% (TEAP, 2019), although Table A4.3 in TEAP (2019) mistakenly stated that 75% was the value used. To further complicate matters, some references have used 0.75% for HFC emissions (IPCC/TEAP, 2005; IPCC, 2006). For this work, because the block-and-pipe market is relatively small for HCFC-141b, what we use for this value is of very small relevance. The largest annual difference in emissions between using 0.75% and 75% is less than 0.7 Gg."**

- Line 267-270: The Weibull lifetime parameters appear fixed across regions. It would be helpful to discuss whether regional differences in product lifespans were considered.

**As stated above, we have added new supplement figures and text discussing regional differences.**

- Line 303: The statement "…, however, globally, most foams are likely not shredded before being landfilled" lacks sufficient literature support. It is recommended that the authors cite appropriate references to substantiate this claim or otherwise clarify that this assertion is based on limited regional evidence or expert judgment.

**Thank you for this comment. We have removed this statement. And please see our response to the next comment for more information about our decommissioning emissions.**

- Line 306-310: The assumed 20% emission rate during dismantling is four times higher than the 5% used in TEAP (2019), which may significantly influence emission and bank estimates. The authors should better justify this choice—ideally with empirical data or sensitivity analysis—and clarify its impact on model outputs.

**We have added additional text to this section discussing the uncertainties associated with this value. We have also reduced it to 15% and now add the statement that the specific choice of this value matters little to our results. The text now states:**

**"… Also, Scheutz et al. (2007) measured an average release of 24% from shredders typical of United States shredding facilities. Key uncertainties regarding how much FBA gets released at and soon after the time of disposal stem from not knowing how much foam is shredded regionally and globally, and of that amount, how finely are the foams shredded. Values for fractional release as time of disposal have ranged from negligible (TEAP, 2019) all the way up to 100%. One hundred percent is almost certainly too much release (TEAP, 2005), and recent estimates have ranged from 2% to 20% (TEAP, 2019). We assume a 15% (s.d., 15%, lognormal distribution) release of the FBA that remains in**

**the decommissioned product to describe losses during the dismantling, transport and disposal processes. This is higher than the 5% used in TEAP (2019) for CFC-11 and much lower than the 100% used in McCullough et al. (2001) for CFC-11. The specific choice of this value, for values described by our assumed uncertainty range, matters little to our comparisons or discussions."**

- Line 335-340: The study assumes independence among most input uncertainties, simplifying the modeling framework. However, market share uncertainties are inherently interdependent, as they must sum to 100%. While a scaling method is applied to address this, the manuscript would benefit from a brief quantitative assessment or supporting citation to demonstrate that this approach does not introduce significant bias.

**We agree with this comment and have added 3 figures to the supplement with a short discussion on biases introduced.**

**In the main text, we added:**

**"This approach leads to a slight low bias of the average market share for moderately-sized markets (Figure S1 and Figure S2). It also leads to a somewhat more substantial low bias in the actual standard deviation of the market size distributions for moderately sized market shares (Figure S3)."**

**And also please see the captions to those new supplementary figures.**

- Line 379: In Figure 6 caption, it may be clearer to describe the top curve as representing the cumulative amount of HCFC-141b that has either been emitted or remains in banks, rather than simply cumulative consumption, to avoid potential confusion.

**We like this suggestion. The beginning of the caption now reads:**

**"Life cycle analysis of all produced HCFC-141b over time. Emission quantities are cumulative over time and banks are instantaneous values. The top of the "production emission" curve represents total HCFC-141b that has been emitted or remains in banks. This is equivalent to cumulative consumption over time, including that which was not reported and was estimated here to be emitted as "Production Emissions", and excluding the small amount of HCFC-141b that is assumed to be captured and destroyed at the time of decommissioning in Europe."**

- Line 505-509: It is suggested that Table A1 in Appendix A be formatted to fit within a single page to enhance readability

**We agree with this comment and hope that it can be done in copy-editing by the journal.**

- Line 510-514: Please ensure that the title of Table A2 appears on the same page as the table for clarity and consistency.

**We have done this.**

- It is also suggested that important sensitivity results (e.g., the relative importance of parameters) be visually summarized in a figure for easier interpretation or listed in Appendix A.

**We have added two figures to the supplement to do this. We have also dropped the color-coding of the table. In Section 3.2, we have now added new text:**

**"Due to the large variation in the sizes of the markets as well as in the magnitudes and the uncertainties ascribed to each parameter, there is a large variation in the impact of the uncertainties of each parameter in Tables 1 and 2 on the emission range shown in the figure. To identify the key uncertainties in the calculations, we have performed Monte Carlo calculations for each parameter individually, with all others fixed. The three most important sources of uncertainty are uncertainties in market segmentation, emissions associated with production (and before sale), and amount of solvent use. Each of these tends to change the entire emission curve roughly proportionately over the time period shown, with other uncertainties demonstrating different temporal impacts on emissions (Figure S4). The top 30 uncertainties when averaged through 2024 are shown in Figure S5 and can provide insight into which parameters should be given the most focus for improving understanding if more confident HCFC-141b emissions are desired. These simulations are driven by the prescribed error on each parameter, and thus, the results are highly dependent on both the estimated parameter values as well as their assigned uncertainties."**

---

## Author Response (AR2)

Our responses are in bold. We very much thank the reviewer for taking another extremely careful look through our manuscript. We agreed with almost all their comments. Please let us know if you have any further questions.

The manuscript reads much better now. However, I still had at some places some problems to follow and would like to ask the authors for a further revision. These are however only minor issues and should be easily be implemented.

Specific comments:

P1, L11 and L12: What is the difference between "Use in emissive applications" and "exist in application and equipment". Isn't that essentially the same?

**We have altered the second sentence of the abstract to make it clear that we are referring to the same applications mentioned in the first sentence.**

P1, L21 and L22: What is an "easily accessible bank" and what is the "total bank"?

**We changed "easily accessible bank" to "easily recoverable portion of the bank" to provide clarity. We hope this also helps with the "total bank" question. We contemplated changing "total" to "entire" but it's not obvious that it would help.**

P2, L44 and L46: Is the abbreviation R/AC really useful? Wouldn't it be better to just omit it and write "refrigeration and air conditioning" a second time?

**We agree since it is never used elsewhere in the manuscript. We have changed it as suggested.**

P3, L83: Instead of just "bottom-up methods" I would suggest to directly write "bottom-up modelling methods".

**We made this change. We also switch the use of "approaches" and "methods". For some reason this sounds better to us.**

P3, L89: Also here I would suggest to write instead of just "methods" rather "modelling methods".

**We made this change (with it now saying "modeling approaches").**

P3, L95: This is why I had the two previous comments, because here you suddenly call it models and not methods.

**This is a good point. Thank you.**

P4, L109: What do you mean with "significantly banked"? Not clear, please rephrase this sentence. Did you mean "mainly" banked?

**We have changed this sentence to read "… unlike datasets for many other ODSs that have significant quantities residing in banks."**

P4, L113: What is the difference between "emissions" and "banks"? Looking at Figure 1 it becomes clearer, thus I would suggest that you already in this paragraph refer to Figure 1.

**In addition to referring to Figure 1 here, we change "calculates banks in" to "calculates the sizes of banks in…"**

P5, L136: Figure should be abbreviated as Fig. and Sections as Sect. unless it appears at the begin of the sentence (Copernicus style). Please correct this throughout the manuscript.

**We have made these changes.**

P7, L160: What is meant with "Parties to the Montreal Protocol"? Parties that participated in or agreed on the Montreal Protocol? Please rephrase sentence to be more precise.

**This sentence now reads "The Montreal Protocol requires that all ratifying countries, which includes all countries in the world, annually report to the Ozone Secretariat the amount of domestic chemical production as well as the amount imported from and exported to other countries for all controlled ODSs.**

P7, L160: report to where? And "imported and exported" quantities not clear in this context. Please rephrase.

**Please also see response to previous statement. In this statement, it states that the reporting is to the Ozone Secretariat. They are the ones that keep all this data.**

P11, Table 2: Units are missing for the lifetime and Weibull scale factor.

**We have added that "s" is unitless and "τ" has units of "yr".**

P17, L401: Which figure? Please be more precise.

**We now explicitly refer to it as Fig. 6.**

P18, L421 and figures: Gg/yr should be Gg yr-1 (Copernicus style).

**We made the change here and in line 469.**

P18, L438 and L443: Instead of writing Wang et al. (2015) parameters, I would suggest to write "parameters provided by" or "parameters provided in".

**We have made these changes.**

P23, L533: Add the reference or table where these are listed.

**We have added references for reported production and published market information and we refer to Tables 1 and 2 for the emissions factors.**

P26, Table A1: What is meant with "but not in the tables"?

**We have changed the 2 instances to "Used in TEAP(2019), but not listed in tables" to hopefully clarify.**

While these values are used in the modeling of that report, the values were not given in the text of the report.

P28, Table A2, Last row and in general: Please give also the numbers or value ranges.

**Since these are the references for the numbers used in Table 2, we would prefer to leave it as it is. Note that the description and range are given in Table 2 in the last row; leaving it as it is is more consistent with the 3 previous rows, too, although we do provide the central value of the emission factor for the inactive bank.**

Note: Currently where is a supplement, but also the same supplement is included in the manuscript. Please take care that in the end not the same figures are published twice.

**Thank you.  We will indeed remove this from the main text.**

Technical corrections:

P1, L19: Add "calculated" so that reads "our calculated emission".

**Done.**

P2, L31: lead to large further climate benefits -> lead to further large climate benefits (?)

**Done.**

P4, L102: Add "modelling" before "approaches".

**Done.**

P4, L120: write "are found" or "are summarized".

**We now say "are provided"**

P5, L130: I would suggest to rather use plural here, thus is -> are and emission -> emissions

**We agree that this sounds better.**

P5, L133: add "the" so that reads "where the consumption".

**Done.**

P7, L162: Add "Ozone" before "Secretariat".

**Done.**

P8, L178 and P9, L226: Figure should not be written in bold text. Use normal text font and abbreviate it a Fig. (Copernicus style).

**Done.**

P9, L205 and L206: UNEP should be written in upright font.

**Done.**

P14, Figure 4 caption, first line: "…….and (3) and……." one "and" is obsolete. Last line repetition of "here".

**We removed the "and" after the "(3)". We have also replaced the second "here" with "in this figure". We feel like if we do not do this, it is unclear whether the statement refers to Figure 1 or Figure 4.**

P15, L341-342: Repetition of "CFC-11"?

**We prefer to retain both "CFC-11" instances to make it clear neither of these references were calculating HCFC-141b.**

P15, L386: Delete "in" and put Sect. 3 in parentheses

**We think the reviewer means line 368 (just transposed the "6" and the "8"). We have made this change.**

P21, Figure 8 caption, L479: Figure 6 should be written in normal text font and abbreviated as Fig. 6.

**Done.**

P21, Figure 8 caption, L480. Panel -> panel

**We made this change in line 479.**

P23, Figure 9 caption, L526: is in same in both panels -> is the same in both panels

**Done.**

P24, L536: Add "bottom-up" so that it reads "The bottom-up model".

**Done.**

**We have made a few additional changes for minor correction, clarity, etc. All changes can be seen in the version submitted with changes tracked. Some of these are described below:**

**In responding to the current set of comments, our abstract has grown to 252 words. We have thus removed "for the period" since it doesn't detract from the point of the sentence and brings us below the 250-word limit.**

**In response to previous reviewer suggestions, we missed two values what should have been updated:**
**Our total bank value for 2030 in line 508 was altered from 1750 Gg to 1700 Gg.**
**Our refrigeration bank in 2030 was altered from 50 Gg to 60 Gg.**

**We have added an additional sentence to the acknowledgements thanking Dr. Velders and the two anonymous reviewers for their helpful comments and suggestions in the process.**

**We have corrected a typo of "or" to "of" and dropped "production" in line 43 of the new Word version just for smoother readability**

**Line 161 of previous version: changed "reported as being aggregated" to "reported in aggregate".**

**L. 179: Since we never discuss individual countries, nor do we show that data, we have removed the example countries in this sentence.**

**L. 246: We change "assume" to "assign" since "assume" has been used multiple times in close proximity.**

**L. 247: we move the statement about emitting installation emissions in the year of production up 2 sentences since it goes better with that thought.**

**Table 2 caption: removed "assumed to be".**

**L 306: We simplified the 2nd term in equation 9.**

**L 342: replaced "for values described by our" to "when considering our".**

**L 343: we added "through the present time" at the end of this sentence for clarification.**

**L 371: wording slightly altered for clarity.**